# Quantitative imaging of transcription in living *Drosophila* embryos reveals the impact of core promoter motifs on promoter state dynamics

Virginia L. Pimmett [1,4], Matthieu Dejean [1,4], Carola Fernandez[1], Antonio Trullo [1], Edouard Bertrand [1,2], Ovidiu Radulescu [3] & Mounia Lagha [1✉]

Genes are expressed in stochastic transcriptional bursts linked to alternating active and inactive promoter states. A major challenge in transcription is understanding how promoter composition dictates bursting, particularly in multicellular organisms. We investigate two key *Drosophila* developmental promoter motifs, the TATA box (TATA) and the Initiator (INR). Using live imaging in *Drosophila* embryos and new computational methods, we demonstrate that bursting occurs on multiple timescales ranging from seconds to minutes. TATA-containing promoters and INR-containing promoters exhibit distinct dynamics, with one or two separate rate-limiting steps respectively. A TATA box is associated with long active states, high rates of polymerase initiation, and short-lived, infrequent inactive states. In contrast, the INR motif leads to two inactive states, one of which relates to promoter-proximal polymerase pausing. Surprisingly, the model suggests pausing is not obligatory, but occurs stochastically for a subset of polymerases. Overall, our results provide a rationale for promoter switching during zygotic genome activation.

[1] Institut de Génétique Moléculaire de Montpellier, Univ Montpellier, CNRS, Montpellier, France. [2] Institut de Génétique Humaine, Univ Montpellier, CNRS, Montpellier, France. [3] Laboratory of Pathogen Host Interactions, Univ Montpellier, CNRS, Montpellier, France. [4] These authors contributed equally: Virginia L. Pimmett, Matthieu Dejean. ✉email: mounia.lagha@igmm.cnrs.fr

In all eukaryotes, transcription of active genes into RNA requires the controlled assembly of multiple protein complexes at promoters[1]. This includes the sequential recruitment of general transcription factors to form the pre-initiation complex (PIC), followed by the recruitment of RNA Polymerase II (Pol II) at the transcriptional start site (TSS)[2–4] of promoters. Among all these complexes, TFIID stands out as the primary core promoter recognition factor that triggers PIC assembly[5]. TFIID includes TATA-binding protein (TBP), which binds upstream of the TSS, as well as 13 TBP-associated factors (TAFs), known to bind downstream promoter elements such as the Initiator element (INR) and the downstream promoter element (DPE) motifs[3,6].

After initiation, Pol II transcribes a short stretch of 30–80 nucleotides before pausing[7]. This step is regulated by the TFIID complex, as directly demonstrated in vitro[8] and inferred from the over-representation of TFIID-bound core promoter motifs (INR, DPE, pause button) in highly paused genes[9–13]. Pausing durations are highly variable among genes[14–16]. However, it is thought that exit from the paused state is a kinetic bottleneck in the transcriptional cycle[17] and is often used as a checkpoint during development to foster coordination in gene activation, plasticity or priming[18–21]. How core-promoter motifs affect this rate-limiting step is still unknown, particularly in the context of a developing embryo.

In parallel to these genomics approaches, single-cell imaging revealed that transcription is not a continuous process over time but occurs through stochastic fluctuations between periods of transcriptional activity and periods of inactivity, called bursting[22]. Direct labeling of newly synthesized RNA with the MS2/MCP amplification system[23,24] has been the method of choice to observe these bursts in real time in living cells[25–27] or multicellular organisms[28,29]. In the context of a developing embryo, these studies revealed how enhancers regulate bursting during pattern formation[30–32]. However, relatively less attention has been given to the impact of core promoter motifs on transcriptional bursting.

To build a mechanistic understanding of bursting beyond a qualitative description of burst size and frequency, it is critical to consider the various timescales of the transcription process and to employ mathematical modeling explicitly describing various promoter states[22]. Indeed, depending on the gene and the cellular context, transcriptional bursting has been shown to occur at multiple timescales[33,34], from seconds (e.g., polymerase clusters, Pol II firing[35–37]), to minutes (e.g., TBP binding/unbinding, transcription factor binding[38,39]) to hours (e.g., nucleosome remodeling, chromatin marks[37,40]). These different timescales can all occur at a single gene, and the term multiscale bursting was coined to describe such complex bursting kinetics[34,37]. This multiscale bursting might explain why the simple two-state model (random telegraph), whereby promoters switch from an active to an inactive state, does not always suffice to reliably capture promoter dynamics[36,37,41,42].

While it is well understood that cis-regulatory sequences must impact transcriptional bursting, with enhancers primarily affecting burst frequencies[43,44], a detailed dissection of the impact of promoter motifs on promoter-state dynamics is lacking. In this study, we sought to examine the role of core promoter sequences on transcriptional bursting in Drosophila embryos. In particular, we focus on two core promoter motifs, the TATA box (TATA-WAWR) and the INR (TCAGTY in Drosophila[6]) which represent pivotal core promoter contact points with the TFIID complex[45,46], and are known to regulate both initiation and promoter pausing[8,12].

The Drosophila early embryo is an ideal system to decipher transcriptional bursting regulation since spatially distinct patterns of gene expression are deployed within a relatively short time frame in a multinucleated syncytium, rapidly dividing, that is highly amenable to quantitative imaging[47]. At this early stage of development, the zygotic genome, initially transcriptionally silent, awakens progressively while cell cycle durations lengthen, a process known as the maternal-to-zygotic genome transition (MZT)[48]. This zygotic genome activation (ZGA) occurs gradually through two waves, a major wave during nuclear cycle 14 (nc14) and a minor wave occurring at earlier stages. Moreover, Drosophila developmental genes show a clear promoter code with well-defined promoter elements[6,49], with differential usage during ZGA[50].

We employed the MS2/MCP system to monitor nascent transcription. We implemented a machine-learning method that deconvolves single-nuclei mRNA production from live Drosophila embryos to detect all single polymerase initiation events. The waiting times between successive initiation events were further analyzed to infer the number of promoter states and the transition rates among these states. Our results show that TATA box-containing promoters are highly permissive to transcription through long ON durations and high Pol II firing rates. However, the presence of an INR in the core promoter necessitates the use of a three-state model, with a second inactive promoter state that is likely a consequence of Pol II pausing. We propose a renewed view of promoter pausing whereby only subsets of polymerases enter into a paused state.

## Results

### A synthetic platform to image promoter dynamics in living embryos.

To examine how the core promoter sequence influences gene expression variation, we developed a synthetic platform whereby several core promoters were isolated, cloned into a minigene, and inserted at the same site in the Drosophila genome (Fig. 1a). This approach allows for a direct comparison of core promoter activity since differences caused by variations in cis-regulatory context, genomic positioning, and mRNA sequence (particularly the 3′UTR) are eliminated. Core promoters were inserted immediately downstream of the snail (sna) distal minimal enhancer (snaE)[51]. The core promoters were selected based on the presence of known core promoter motifs[49], such as a TATA box in sna or an INR in kruppel (kr) and Insulin-like peptide 4 (Ilp4) (Fig. 1a, Supplementary Fig. 1, and Supplementary Movies 1 and 5). We also selected the brinker (brk) core promoter, which is devoid of any known canonical promoter motifs. None of these four promoters have a DPE, however, Ilp4 possesses a bi-partite bridge element[49]. These developmental genes are endogenously expressed during nc14[52,53] and precise TSS positions were established using embryonic CAGE (Cap Analysis of Gene Expression) datasets[54] (Supplementary Fig. 1). For all transgenes, we used 100 bp of a promoter sequence, as previous work established that minimal promoter sequences are sufficient to establish pausing in vivo[12,21].

To track transcription, 24 MS2 stem loops were placed in the 5′ UTR downstream of the promoter, followed by the sequence of the Drosophila yellow gene, a gene fragment used as a reporter because of its large size and lack of endogenous expression in the early embryo[55]. MS2 stem loops in transcribed mRNA were bound by a maternally supplied MCP-GFP fusion protein, which allowed the detection of transcriptional foci as bright GFP spots. We imaged living Drosophila embryos at a high temporal resolution of one frame per 3.86 s and focused on the first 30 min of nc14[56]. To ensure that all quantified nuclei experience similar peak levels of transcriptional activators, we restricted our analysis to a defined domain within the mesoderm (25 μm on either side of the presumptive ventral furrow; Fig. 1b)[57]. Signal intensity at the transcription site was retrieved for each nucleus after 3D detection and tracked throughout nc14 (with mitosis between nc13 and nc14 considered as time zero; see "Methods"

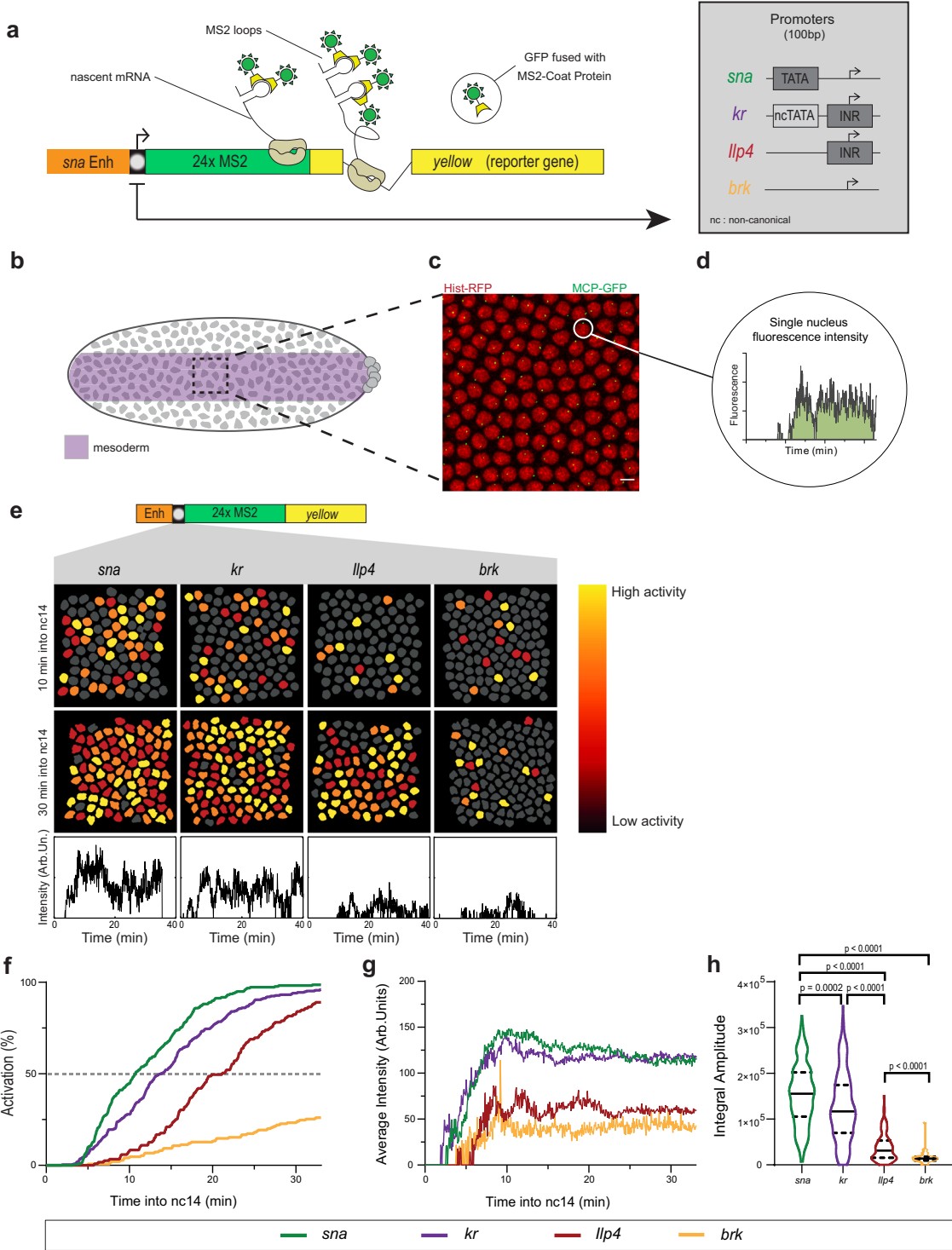

section) and then associated with their nearest nucleus. This generated individual 4D nuclear trajectories (Supplementary Fig. 2; "Methods").

**Core promoters differentially affect transcriptional synchrony.** We hypothesized that differences in core promoter motifs would result in variability in gene expression independently of specific enhancer contacts. To test this, we characterized single-nucleus transcriptional activity (Fig. 1c–e), as well as the initial timing of activation. As previously reported with fixed sample approaches[21,58], we observed a spectrum of synchrony profiles

(Fig. 1f), defined as the temporal coordination of activation among a spatially defined domain. The TATA-driven *sna* and INR-driven *kr* minimal promoters showed a rapid activation with a t50 (the time needed for 50% of nuclei within the region of interest to show GFP puncta) of 11 min and 14 min, respectively while the INR-driven *Ilp4* promoter was delayed with t50 of 21 min. The *brk* core promoter did not surpass 24% of nuclei activated throughout nc14.

In addition to differences in synchrony, the overall mRNA production also varied between transgenes (Fig. 1g). Within our four developmental promoters, promoters showing more rapid reactivation in nc14 also had higher TS intensity in active nuclei,

**Fig. 1 A synthetic transgenic platform to image promoter dynamics. a** Schematic view of transgenes used to study transcriptional dynamics of *sna*, *kr*, *llp4*, and *brk* core promoters. A minimal *sna* enhancer was placed upstream of the core promoter followed by 24xMS2 repeats and a *yellow* reporter gene. Core promoter motifs are indicated in the inset. **b** Schematic of *Drosophila* embryo showing spatial restriction of analysis to presumptive mesoderm (purple). **c** Maximum intensity projection of representative 15 μm Z-stack of *snaE < snaPr < 24xMS2-y* (*snaE < sna*) nc14 embryo showing MS2/MCP-GFP-bound transcriptional foci (GFP) and nuclei (His-RFP). Scale bar is 5 μm. **d** Sample single-nuclei trace showing GFP fluorescence during nc14. The surface of the green region indicates trace integral amplitude. **e** False-colored frames from live imaging of indicated promoters showing relative instantaneous fluorescence intensity in early and late nc14. Inactive nuclei are gray and highly active nuclei are in yellow. **f** Cumulative activation curves of all nuclei during the first 30 min of nc14 for *sna* (green), *kr* (purple), *llp4* (red), and *brk* (blue). Time zero is from anaphase during nc13-nc14 mitosis. **g** Average instantaneous fluorescence of transcriptional foci of active nuclei during the first 30 min of nc14 for *sna* (green), *kr* (purple), *llp4* (red), and *brk* (blue). Time zero is from anaphase during nc13-nc14 mitosis. **h** Distribution of individual trace integral amplitudes from first 30 min of nc14 for *sna* (green), *kr* (purple), *llp4* (red), and *brk* (blue). The intensity amplitude at a given time may result from the overlap of several bursts and is a convolution of promoter active/inactive times, polymerase initiation frequency, and the duration of a single polymerase signal. The integral amplitude estimates the transcriptional activity and the total number of transcripts at a steady state; it is proportional to the probability of active state (pON) initiation rate ($k_{INI}$) and to the duration of the signal. Solid lines represent median and dashed lines first and third quartiles, using a one-tailed Kruskal–Wallis test for significance with multiple comparison adjustments. Statistics: *snaE < sna*, 216 nuclei, 3 movies; *snaE < kr*, 243 nuclei, 4 movies; *snaE < llp4*, 114 nuclei, 2 movies; *snaE < brk*, 45 nuclei, 2 movies. See Supplementary Movies 1 and 5.

indicating increased instantaneous mRNA production. To examine the total mRNA output, we looked to the integral amplitude, or the area of the curve defined by the average MCP-GFP signal plotted over time (Fig. 1d). The integral amplitude showed that promoters mediating faster activation and higher instantaneous mRNA production had a higher total mRNA output (Fig. 1h).

Thus, we established an experimental setup and quantification pipeline that allows disentangling the contribution of minimal promoter sequences on transcriptional activation of individual, naturally synchronized nuclei within a well-defined, homogeneous spatial domain in live embryos. This initial quantitative comparison of four natural developmental promoters suggests that those with a canonical promoter motif (either TATA or INR) tend to produce higher levels of expression.

**A machine-learning method to infer promoter-state transition rates.** With current labeling and imaging technology, individual transcriptional initiation events cannot be easily and directly visualized, tracked, and quantified in the early *Drosophila* embryo. Live imaging of transcription typically shows spots comprised of several newly synthesized mRNAs resulting from the action of multiple polymerases. In order to calibrate fluorescent signals from live imaging, we used single-molecule hybridization experiments[28]. Using the fluorescence of a single mRNA molecule, we estimated the average number of mRNA molecules present at the TS within the nucleus at a steady state (Supplementary Fig. 3). This calibration step allowed us to express single-nuclei TS intensities as an absolute number of transcribing polymerases (Fig. 2a).

We then used a recently described novel machine-learning method to infer the transcriptional bursting mechanism[59]. This method involves three major steps: detection of successive initiation events for each nucleus, multiexponential parametric regression of the distribution of waiting times between successive events, and identification of Markovian promoter-state transition models.

In order to detect initiation events, we considered that each trace results from the convolution between (i) the sequence of initiation events marked by a rise in GFP intensity plotted over time, and (ii) the signal produced by a single polymerase (Fig. 2b)[33,59]. The deconvolution procedure uses a genetic algorithm to determine optimal Pol II positioning within the gene body (Fig. 2c) and thus the time between successive Pol II initiation events (Δt) (Fig. 2d). In this first step, some specific parameters are fixed, such as the temporal resolution (3.86 s), the speed of Pol II elongation (45 bp s$^{-1}$ in *Drosophila* embryos[60]), the size of the transcript (5.6 kb), and the retention time of the

mRNA at the transcription site (assumed to be small relative to the time required to produce a transcript).

Having obtained temporal maps of individual Pol II transcription events, the method then statistically analyzes the distribution of waiting times between two successive polymerases (Δt)[59]. A multiexponential regression fitting was applied to the distribution of Δt, indicating the number of rate-limiting transitions required to best fit the data ("Methods"). We used confidence intervals and a Kolmogorov–Smirnov test to rigorously determine the smallest number of rate-limiting steps fitting our experimental data (Supplementary Table 1). Following the principle of parsimony and to avoid overfitting, models with a larger number of steps and more parameters were not retained even if they also fit well. Importantly, this step uncovered the number of characteristic timescales of transcriptional fluctuations that are tantamount to the number of promoter states.

Finally, a transcriptional model of the promoter was established, with multiple states and timescales inferred from the parameters of the multiexponential Δt distribution (Supplementary Table 2). This step permitted the estimation of the kinetic parameters such as the $k_{ON}$ (rate of switching from a transcriptionally nonpermissive state to a permissive one), $k_{OFF}$ (switching rate from a transcriptionally permissive to a nonpermissive state), and $k_{INI}$ (the rate of Pol II initiation events once the promoter is in a transcriptionally permissive state) for the simplest two-state model. However, kinetic estimates of more complex models (three or more promoter states) can also be derived from the Δt distribution. The accuracy and robustness of the deconvolution method were tested by applying it to artificial data with known positions of transcription initiation events and known kinetic parameters (Fig. 2e–h). These artificial data were generated using the Gillespie algorithm and proved the robustness of the approach (see "Methods").

Importantly, this approach provides a time map of transcription events in a cell population in a model-independent manner. The number of promoter states is evaluated during the multiexponential fit procedure, but with no a priori constraints on the result[59]. This contrasts with current methods which directly fit a particular transcription model to the data, such as methods based on the autocorrelation functions[61–63], maximal likelihood estimates[36], or Bayesian inference[30,64,65].

**The *sna* promoter as a model to investigate the impact of core promoter on transcriptional dynamics.** To investigate the quantitative dynamics of the *sna* promoter, we applied this method to hundreds of nuclei pooled from *snaE < sna* embryos. A heatmap of positions of single initiation events for each nucleus

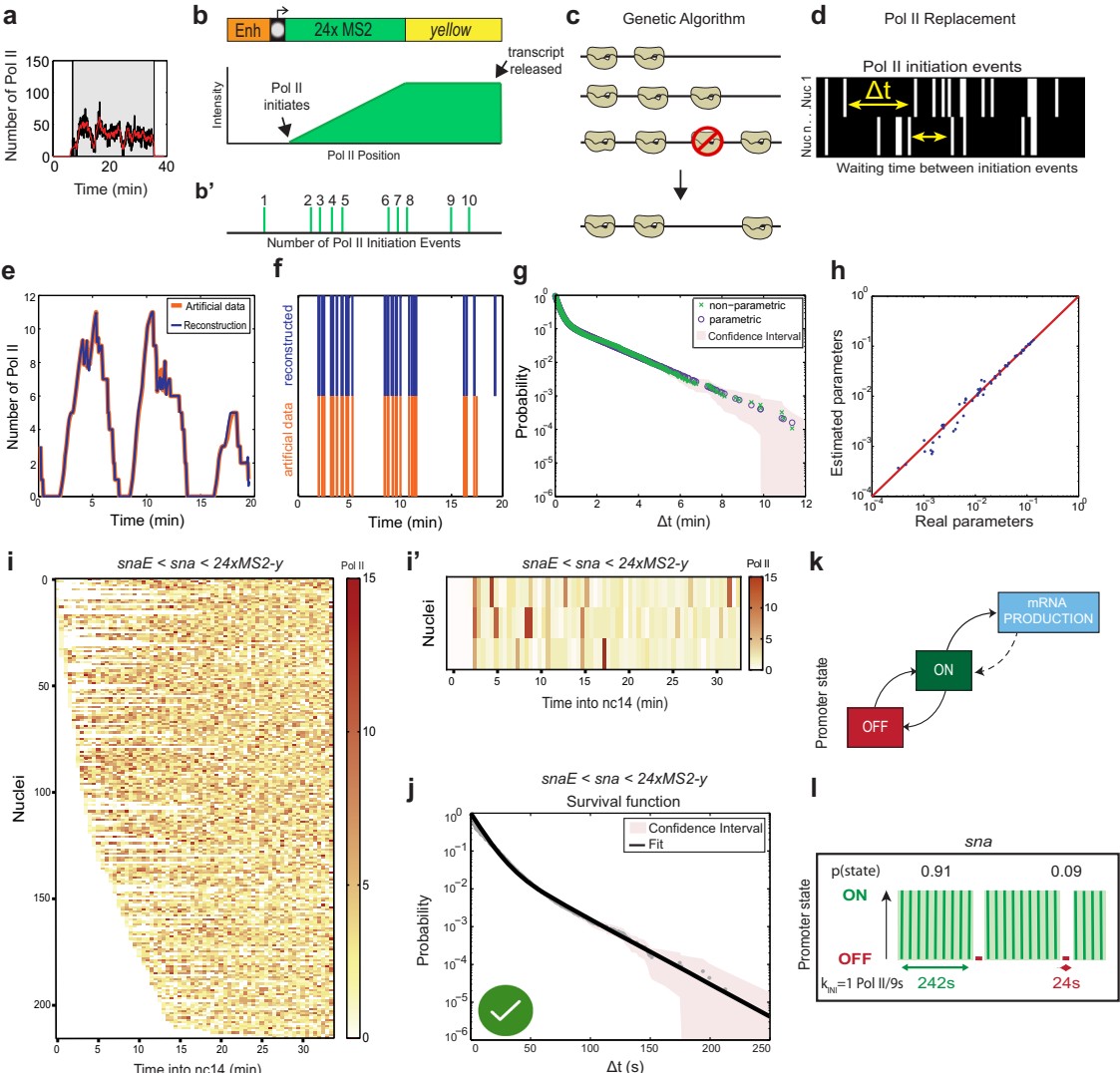

**Fig. 2 From live imaging of transcription to positions of initiation events, a machine-learning procedure. a** Representative trace of a single-nucleus transcriptional activity. The gray box indicates the analyzed transcriptional window, the black curve represents the signal expressed as the number of polymerases, and the red curve the reconstructed signal after the deconvolution procedure. **b** Fluorescence intensity at the transcription site is a function of the passage of a single polymerase as well as the number of polymerase initiation events. **c** A genetic algorithm is used to decompose fluorescence intensities and optimally locate polymerases within the gene body. **d** For each nucleus, the position of polymerases is used to extract Δt, the lag time between two successive initiation events. **e** Simulated representative trace (orange) and the number of polymerases after deconvolution (blue). **f** Known positions of polymerases (orange) and reconstructed polymerase positioning after deconvolution. **g** Nonparametric (green) and parametric (blue) survival function estimate for simulated data estimated using the Kaplan–Meyer method. The shaded region indicates 95% confidence interval estimated based on Greenwood's formula. **h** Comparison of known parameters used to generate artificial data (real parameter) with parameters obtained by applying our analysis pipeline (estimated parameters). Red line indicates perfect concordance. **i** Heatmap showing the number of polymerases for the 216 *snaE < sna* nuclei as a function of time. Each row represents one nucleus, and the number of Pol II initiation events per 30 s bin is indicated by the bin color. **j** A representative population from **i**. **k** Survival function of the distribution of waiting times between polymerase initiation events (red circles) and the two-exponential fitting of the population estimated using the Kaplan–Meyer method (black line). The shaded region indicates 95% confidence interval estimated based on Greenwood's formula (see Supplementary Table 1). A green check indicates accepted fitting. **l** A two-state model showing the probabilities to be in either the permissive ON state or the inactive OFF state for the *snaE < sna* transgene. **m** Representation of estimated bursting dynamics for the *snaE < sna* transgene. Permissive ON state durations are depicted in green and inactive OFF states in red, and probabilities of each state shown above (see also Supplementary Table 2). Statistics: *snaE < sna*, 216 nuclei, 3 movies; see Supplementary Movie 1.

revealed the *sna* promoter drove frequent transcriptional initiation events. Silent periods with no initiating polymerase were very rare (Fig. 2i–j, white bars), consistent with the results obtained from fixed embryos[21,66]. We found that a biexponential (sum of two-exponential functions) fitting correctly fit the survival function of polymerase waiting times (Fig. 2k). The biexponential fitting is only compatible with a two-state model (Fig. 2k). Thus, the transcriptional activity of the *sna* promoter can be described

by the random telegraph model, with a simple random switch between an inactive OFF and a permissive ON state, from which transcription initiates at a given rate ($k_{INI}$; Fig. 2l). The probability to occupy the permissive ON state was high in the case of *sna* (0.91; Fig. 2m). The $T_{ON}$ was estimated to be 242 s, with a $T_{OFF}$ of 24 s. The $k_{INI}$ of *sna* promoter was estimated at one initiation event every 9 s (Fig. 2m), consistent with the inferred initiation rate at its endogenous locus[66] and in the range of recent

estimates of initiation frequencies for other developmental genes in *Drosophila*[32] and in mammalian cells[37]. Collectively this shows that we can locate individual initiation events, determine the number of promoter states, select a promoter model by the principle of parsimony, and then estimate average promoter switching rates in vivo.

**The TATA box regulates the ON and OFF duration but not the number of states**. To determine how the TATA box influences transcriptional kinetics in vivo, we developed a series of *sna* core promoter mutants (Fig. 3a) where the TATA box was replaced with either the TATA-like sequence of *kr* (*snaTATAlight*) that is bound by TBP[50] (Supplementary Fig. 1) or a non-TATA sequence where the first four bases are mutated (*snaTATAmut*). Surprisingly, the strong TATA mutation did not completely abolish the activity of the *sna* promoter, although this mutant promoter was devoid of any known core promoter motifs. In comparison to the *sna* promoter, the synchrony of the *sna* TATA mutants was reduced in a graded manner relative to the fidelity of the TATA box (Fig. 3b, Supplementary Movies 1–3). The *sna* promoter reached synchrony at ~11 min into nc14, while the *snaTATAlight* promoter reached synchrony at 24 min and the *snaTATAmut* promoter never reached synchrony. This implies that the TATA box influences synchrony of activation, most likely by promoting and stabilizing TBP binding (Supplementary Fig. 1).

Similar to the activation profiles, instantaneous TS intensities were directly correlated with TATA box sequence fidelity (Fig. 3c). Interestingly, while both the *sna* promoter and *snaTATAlight* promoters appeared to reach a steady state, the *snaTATAmut* was unable to do so. This may have been the result of MCP-GFP signal being too low to consistently surpass the detection threshold, as single-molecule FISH experiments indicated most nuclei in the ventral region were at least weakly active in nc14 *snaTATAmut* embryos (Supplementary Fig. 4a–c). The total mRNA production of each promoter measured by the integral amplitude was similarly affected (Fig. 3d), with the *sna* promoter driving the highest total mRNA expression while the TATA box mutants expressed reduced amounts of mRNA correlated to the fidelity of the TATA box.

We next asked whether the *snaTATAlight* and *snaTATAmut* transgenes still followed a two-state model. For each nucleus of each of these three transgenic genotypes, we located single Pol II initiation events during the first 30 min of nc14 (Fig. 4a–f and Supplementary Fig. 4d–f). By examining the survival curve of Pol II waiting times, the dynamics of the *sna* TATA mutants appeared to be accurately recapitulated by the two-state random telegraph model, similarly to the unmutated *sna* promoter (Supplementary Fig. 4g–i, Supplementary Table 1). We then analyzed which kinetic parameters changed in the mutant promoters. Extracting kinetic parameters for the *snaTATAlight* transgene revealed a relatively mild effect on initiation rates (1 Pol II per 13 s for *snaTATAlight* instead of 1 Pol II per 9 s) (Fig. 4g–i and Supplementary Table 2). In contrast, we observed a strong increase in the $T_{OFF}$ (Fig. 4g–k) for the *snaTATAlight* promoter relative to the *sna* promoter (Fig. 4g–i, k and Supplementary Fig. 5b). The strong TATA mutation led to a similar trend, although due to the weak activity of this promoter, the number of analyzed nuclei for this mutant is lower than for the other promoters. The kinetic parameters of TATA mutant promoters led to an overall reduction in burst size, defined as the number of transcripts produced during an active period (Fig. 4l). The reduction was primarily due to a decrease in the duration of the ON periods, consistent with a destabilized TBP/TATA box interaction or reduced TBP recruitment to the promoter.

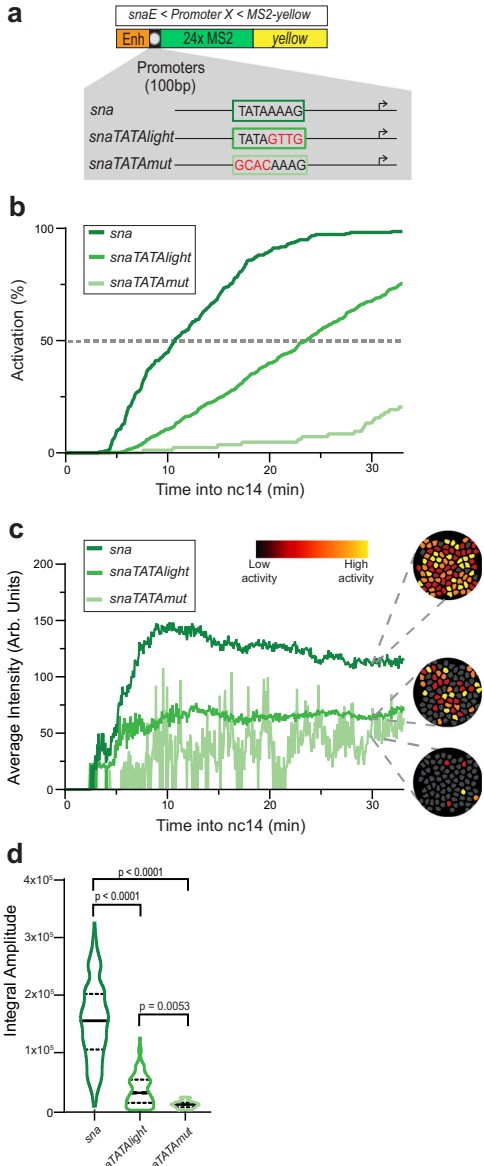

**Fig. 3 Decoding the role of the TATA box on promoter dynamics. a** TATA box mutations of the *sna* promoter. The *snaTATAlight* mutation corresponds to the sequence of the non-canonical TATA box from the *kr* promoter. **b** Cumulative active nuclei percentage within the mesodermal domain for *sna* (dark green), *snaTATAlight* (green), and *snaTATAmut* (light green). **c** Average instantaneous fluorescence of transcriptional foci of active nuclei during the first 30 min of nc14 for *sna* (dark green), *snaTATAlight* (green), and *snaTATAmut* (light green). Time zero is from anaphase during nc13-nc14 mitosis. False-colored panels on the right are colored according to instantaneous fluorescence intensity, with inactive nuclei shown in gray and highly active nuclei in yellow. **d** Distribution of individual trace integral amplitudes from first 30 min of nc14. The solid line represents median and dashed lines the first and third quartiles, using a one-tailed Kruskal–Wallis test for significance with multiple comparison adjustments. Statistics: *snaE < sna*, 216 nuclei, 3 movies; *snaE < snaTATAlight*, 353 nuclei, 6 movies; *snaE < snaTATAmut*, 21 nuclei, 3 movies. See Supplementary Movies 1–3.

The impact of TATA on the duration of ON/OFF promoter states observed in *Drosophila* embryos is also in agreement with results from live imaging of *HIV-1* transcription in human cells[37], and with genomic studies in cultured mammalian cells, where TATA-driven promoters are associated with large burst sizes[67]. They differ from those obtained with mutations of the native

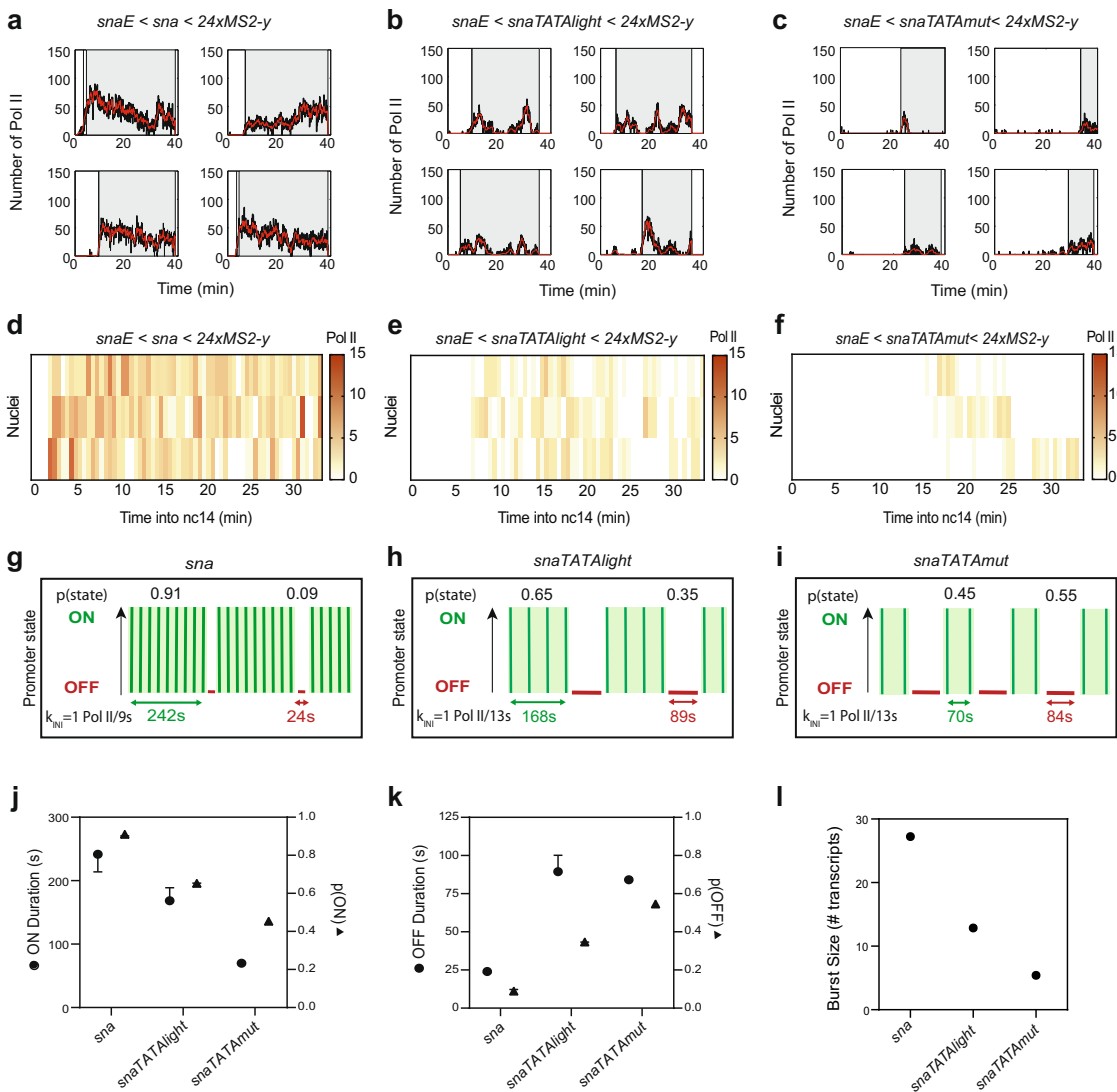

**Fig. 4 TATA box motif leads to long durations of a permissive promoter state. a–c** Representative traces of single-nuclei transcriptional activities for the indicated genotypes. Gray boxes indicate the analyzed transcriptional windows, black curves represent the signal expressed as the number of polymerases, and red curves represent the reconstructed signal after the deconvolution procedure. **d–f** Heatmaps indicating the number of polymerases during nc14 for each genotype. **g–i** Representation of estimated bursting parameters for *sna* (**g**), *snaTATAlight* (**h**), and *snaTATAmut* (**i**) promoters (see also Supplementary Table 1 and Supplementary Table 2). **j** Duration and probability of the permissive ON state. Error bars represent the smallest and largest values in optimal and close to optimal solutions (see "Methods"). **k** Duration and probability of the OFF state. Error bars represent the smallest and largest values in optimal and close to optimal solutions (see "Methods"). **l** Estimated burst size calculated as $k_{INI}/k_{OFF}$[75] using the optimal kinetic parameters (Supplementary Table 2). Statistics: *snaE < sna*, 216 nuclei, 3 movies; *snaE < snaTATAlight*, 353 nuclei, 6 movies; *snaE < snaTATAmut*, 21 nuclei, 3 movies. See Supplementary Movies 1–3 and Supplementary Table 2.

*actin* promoter in *Dictyostelium*[36], which may be due to differential behavior between induced genes and constitutive genes or to compensation by other genomic elements[36].

We conclude that in *Drosophila* embryos, the TATA box largely controls gene expression through lengthening the ON state duration at the expense of the OFF state duration, but does not alter the number of promoter states.

**The INR motif induces a third rate-limiting promoter state.** At the biochemical level, transcriptional activation is a multistep process requiring the orchestration of many factors[3], and yet TATA-driven promoters can be modeled with a simple two-state model corresponding to one kinetic rate-limiting step in this complex process. We hypothesized that additional rate-limiting steps were likely to exist and could be identified using

our imaging-based methodology. As the TATA box did not affect the number of promoter states for the *sna* promoter, we examined the effect of another core promoter motif, the INR. In *Drosophila* cells and embryos, the INR is associated with stably paused genes, genome wide[7,9,15,68]. Moreover, analysis in cell culture indicated that genes with increased pausing stability tend to harbor an INR motif in the core promoter, and in particular a preferential G at the +2 position[12]. We, therefore, reasoned that manipulating the INR motif in embryos might affect pausing and therefore may induce changes in rate-limiting promoter states.

To examine the role of the INR core promoter motif, we created a series of transgenes in which we manipulated the INR without changing the enhancer or the downstream gene sequence (Fig. 5a). The *sna* core promoter does not have an INR, while the core promoter of *kr* has a natural INR sequence with a G at +2 and

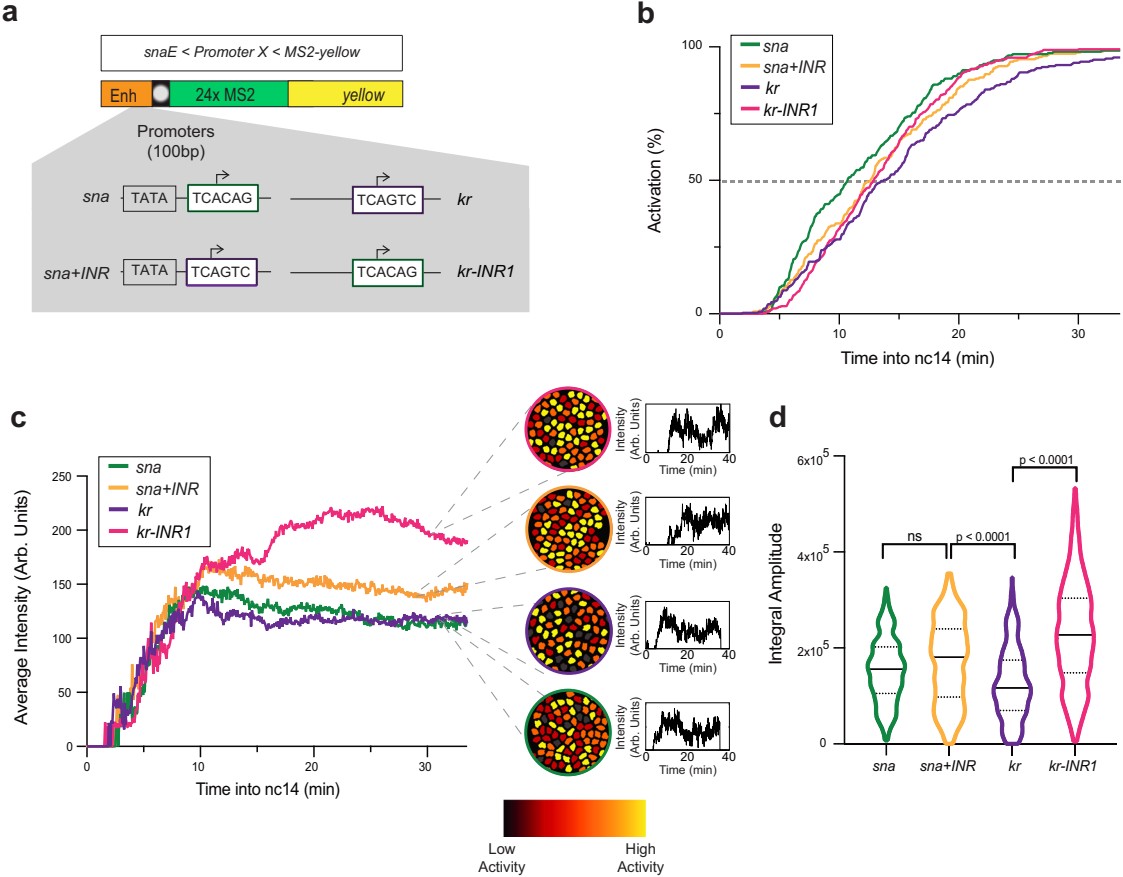

**Fig. 5 Decoding the role of the Initiator motif on promoter dynamics. a** Schematic of the transgenes used to decipher the impact of the INR motif. The *sna* core promoter has the TSS region replaced with the INR of *kr* (*sna+INR*), and the *kr* core promoter INR motif is replaced with the TSS region of *sna* (*kr-INR1*). **b** Cumulative active nuclei percentage within the mesodermal domain for *sna* (green), *sna+INR* (yellow), *kr* (purple), and *kr-INR1* (pink). **c** Average instantaneous fluorescence of transcriptional foci of active nuclei during the first 30 min of nc14 for *sna* (green), *sna+INR* (yellow), *kr* (purple), and *kr-INR1* (pink). Time zero is from anaphase during nc13-nc14 mitosis. False-colored panels on right are colored according to instantaneous fluorescence intensity with inactive nuclei in gray and highly active nuclei in yellow. **d** Distribution of individual trace integral amplitudes from first 30 min of nc14 for *sna* (green), *sna+INR* (yellow), *kr* (purple), and *kr-INR1* (pink). The solid line represents the median, and dashed lines represent the first and third quartiles, using a one-tailed Kruskal–Wallis test for significance with multiple comparison adjustments. Statistics: *snaE < sna*, 216 nuclei, 3 movies; *snaE < sna+INR*, 236 nuclei, 4 movies; *snaE < kr*, 243 nuclei, 4 movies; *snaE < kr-INR1*, 342 nuclei, 5 movies. See Supplementary Movies 1, 4–6.

is stably paused[16,18]. Moreover, the *kr* promoter possesses a non-canonical TATA box (*TATAlight*, Fig. 1a). We exchanged the TSS region of *sna* with the INR of the *kr* promoter (*sna+INR*). We also created a transgene (Fig. 5a) whereby the INR of *kr* was replaced with the TSS region of *sna* (*kr-INR1*). Interestingly, neither loss of the INR in the *kr-INR1* transgene nor gain in the *sna+INR* transgene strongly affected synchrony (Fig. 5b and Supplementary Movies 1, 4, 5). The addition of an INR to *sna* did not dramatically affect mRNA output (Fig. 5d). Both the *sna+INR* and *kr-INR1* transgenes had higher instantaneous activity than the cognate wild-type promoters (Fig. 5c). Moreover, a second mutation of the INR in *kr* (*kr-INR2*) had similar synchrony and mRNA production (average instantaneous intensity) as in the wild-type *kr* promoter (Supplementary Fig. 6a–c). We then applied the deconvolution procedure to each genotype (Fig. 6a and Supplementary Figs. 6d–k and 7a, d–k). Surprisingly, our analysis of polymerase initiation events indicated that, for *sna+INR* nuclei, a two-state model was not sufficient to fit the data (Fig. 6a, b, Supplementary Fig. 7i, and Supplementary Table 1). Instead, the *sna+INR* survival function was well fitted by adding an extra exponential term. Thus, a three-state model appropriately recapitulated *sna+INR* promoter dynamics (Fig. 6a, b and Supplementary Fig. 7i–i'). Similarly, the *kr* promoter also required a three-exponential fitting (Fig. 6b and

Supplementary Fig. 6h–h'). In contrast, removing the INR from this promoter (*kr-INR1* promoter) led to a two-exponential fitting for the survival function (Fig. 6b and Supplementary Fig. 6i–i'). Similar results were obtained with a second INR mutant (*kr-INR2*) (Supplementary Fig. 6j–j'). Furthermore, the natural INR-driven *Ilp4* promoter was also associated with a three-exponential fitting, and mutation of its INR sequence (*Ilp4-INR*) resulted in a reversion to a two-exponential fit (Fig. 6b and Supplementary Fig. 8d, e, h, i'). These results, therefore, suggest that the INR motif is associated with a third rate-limiting promoter state.

**Pol II pausing is associated with a third promoter state.** Given the correlation between the presence of an INR motif and the number of characteristic timescales and thus the number of promoter states (three in the presence of an INR), we reasoned that one of these states could be polymerase pausing. To test this hypothesis, we impaired the establishment of pausing by reducing the expression of the largest subunit of the pause-inducing negative elongation factor complex (NELF), NELF-A. Previous work demonstrated that depletion of NELF by RNAi globally reduced promoter-proximal pausing in *Drosophila* cells[13], and that loss of the human C-terminal NELF-A "tentacle" blocked stabilization of pausing[69]. We combined

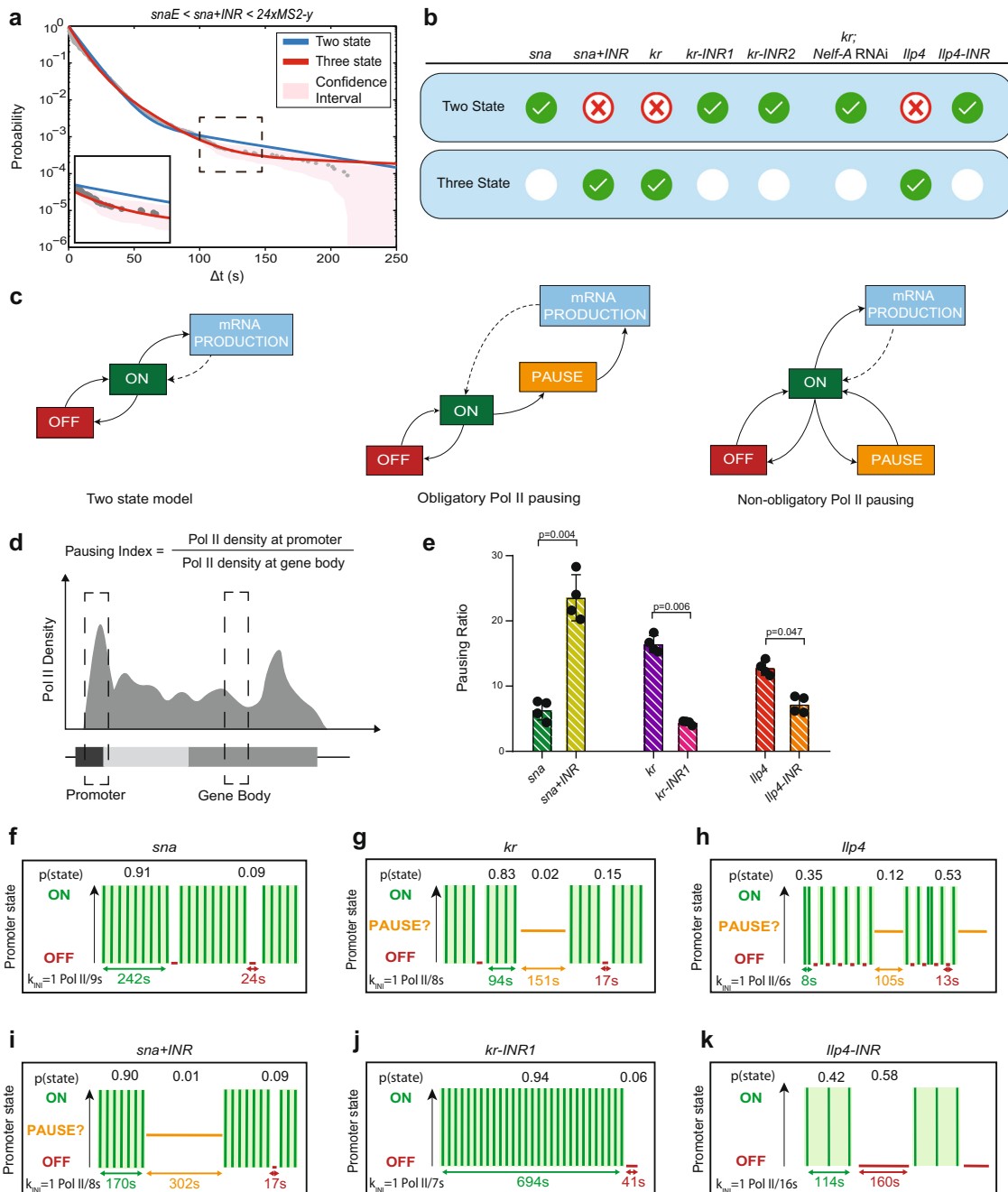

**Fig. 6 The INR motif induces an extra promoter state related to pausing. a** Survival function of the distribution of waiting times between polymerase initiation events (gray circles). Two-exponential fitting of the population (blue line), and three-exponential fitting (red line) estimated using the Kaplan–Meyer method. The shaded region represents 95% confidence interval estimated based on Greenwood's formula (see also Supplementary Table 1). **b** Table indicating the most parsimonious number of states required to fit each indicated genotype (see also Supplementary Table 1). A green check indicates an accepted fitting. **c** Schema of the two-state telegraph model, a three-state model with obligatory pausing, and a three-state model with non-obligatory pausing. **d** Pausing index calculation. **e** Pausing index of indicated transgenes measured by ChIP-qPCR (average + SD), $n = 4$ biological replicates with a Mann–Whitney test. **f**–**k** Representation of *sna*, *sna+INR*, *kr*, *kr-inr1*, *Ilp4*, and *Ilp4-INR* estimated bursting dynamics. Permissive ON states are in green, inactive PAUSE states in orange, and inactive OFF states in red. Statistics: *snaE < sna*, 216 nuclei, 3 movies; *snaE < sna+INR*, 236 nuclei, 4 movies; *snaE < kr*, 243 nuclei, 4 movies; *snaE < kr-INR1*, 342 nuclei, 5 movies, *snaE < snaTATAlight*, 353 nuclei, 6 movies; *snaE < snaTATAlight+INR*, 193 nuclei, 3 movies; *snaE < Ilp4*, 114 nuclei, 2 movies; *snaE < Ilp4-INR*, 48 nuclei, 2 movies. See Supplementary Movies 1, 4–6, Supplementary Table 2.

the inducible RNAi/GAL4 system with our MS2/MCP-GFP reporter to examine the transcriptional dynamics of the three-state *kr* promoter. We performed live imaging of the *kr*-MS2 transgene and compared a control *white* RNAi knockdown with embryos harboring a maternal depletion of *Nelf-A* (evaluated to >85% knockdown by RT-qPCR, Supplementary Fig. 9a, Supplementary Movies 7 and 8). Remarkably, reducing *Nelf-A* transcript levels led to a change in *kr*

promoter dynamics, with a reversion to a two-promoter-state dynamic and much less frequent long waiting times (Fig. 6b and Supplementary Fig. 9e-f', arrowheads, Supplementary Table 1). These data, combined with the effect of INR mutations in *cis*, led us to propose that one of the two inactive promoter states observed with the three-state promoters could be associated with promoter-proximal polymerase pausing.

To confirm that the INR motif was truly altering Pol II pausing, we performed Pol II ChIP-qPCR on our promoter transgenes from staged embryos. We evaluated the strength of pausing by computing the pausing index. The pausing index (PI) was obtained by quantifying the Pol II binding at the promoter relative to the gene body[70] (Fig. 6d). As shown previously, in our transgenic embryonic extracts encompassing various stages from nc13 to late nc14, the wild-type *sna* promoter is paused[18,21,50], but the addition of an INR motif significantly increased its PI (Fig. 6e), consistent with INR-mediated lengthening of the pause duration observed in cell culture[12]. To examine the inverse scenario, we also quantified the PI of the *kr* and *kr-INR1* transgenic promoters. Similar to what is observed in the endogenous promoter[18], the *kr* transgene was highly paused (Fig. 6e). Mutation of the INR in the *kr-INR1* transgene reduced pausing (Fig. 6e). Likewise, the *Ilp4* promoter from our transgene showed a high level of pausing, which decreased upon mutation of its INR sequence (Fig. 6d). Both the *sna+INR* and *kr* promoters also show increased NELF-E enrichment at these promoters by ChIP-qPCR on embryonic extracts (Supplementary Fig. 9g). Taken together, characterization of the INR core promoter motif suggests the presence of a strong INR motif results in stabilization of pausing in vivo.

**Pol II does not undergo systematic pausing.** Next, we envisaged how pausing could be modeled by three distinct promoter states. The current view of pausing is that all polymerases systematically enter into pausing, a step followed by either a productive elongation or a termination[17,71–73]. We therefore initially tested if our data could be explained by a model consisting of three promoter states (inactive OFF, permissive ON, and paused) where all productive polymerases undergo pausing prior to transitioning to the permissive ON state (Fig. 6c). This model was clearly incompatible with our data, as the fitting exceeded the bounds of the 95% confidence interval (Supplementary Figs. 6l and 7i''). Recent work on the prototypic model of a highly paused promoter, the *HIV-1* promoter, also found that modeling transcriptional dynamics with obligatory pausing was not in agreement with live-imaging data in HeLa cells, and instead proposed an alternative model of pausing where only a fraction of polymerases are subject to pausing in a stochastic manner[59]. We asked if this alternative non-obligatory pausing model could be applicable to our findings (Fig. 6c). According to the goodness of fit, nonsystematic pausing was compatible (Supplementary Figs. 6h' and 7i'). Our machine-learning method enabled the estimation of the kinetic transition rates between these states, summarized in Supplementary Table 2 and Supplementary Fig. 5a–c.

We conclude the presence of an INR motif translates into a longer paused state that creates an additional rate-limiting step during transcription in vivo. In our kinetic model, the inactive states could in principle equally correspond to pausing. These two-promoter states are discernible by their distinct timescales, on the order of seconds and minutes, respectively (Supplementary Table 2). Because the NELF knockdown dramatically decreases the frequency of long waiting times (Supplementary Fig. 9e–f), we favor the longer-lived inactive state, which lasts on the order of minutes, as the potential pausing state. The second short-lived inactive state, lasting on the order of seconds and present for both TATA- and INR-containing promoters, would then correspond to a nonpermissive promoter state, possibly TFIID-unbound as its frequency increases in TATA mutants (Fig. 4k and Supplementary Table 2). As the *sna* and *kr* endogenous promoters drive a high level of gene expression (Supplementary Fig. 1), it is reasonable to hypothesize that such a nonpermissive state is transient and infrequent (Supplementary Table 2).

**INR control of promoter switching kinetics.** Like the natural *sna* promoter, the *sna+INR* promoter showed a high probability of occupying the transcriptionally permissive ON state, with a long lifetime of 170 s. This ON duration was slightly reduced compared to the *sna* transgene (Fig. 6f, i), possibly implying competition between the natural canonical TATA box and the added INR in the *sna+INR* combination (see below). The major difference between these two promoters was the existence of two distinguishable inactive states linked to the presence of the INR sequence. One of these was short-lived (17 s) and one long-lived (302 s) (Fig. 6f, i, Supplementary Fig. 5b, c, and Supplementary Table 2). The stable long-lived paused state was not observed with the *sna* promoter but did occur in the presence of the INR-containing *kr* promoter.

In the case of *kr* and *Ilp4* developmental promoters, the paused state lasted ~151 and 105 s, respectively, but was reached relatively rarely (Fig. 6g, j, Supplementary Fig. 5c, and Supplementary Table 2). Pausing was not observed upon two types of mutations of the *kr* INR motif (Fig. 6j, Supplementary Fig. 6i–j', m, and Supplementary Table 2). Thus our data favor a model where the paused state lasts several minutes (in the case of *kr*, *sna + INR* and *Ilp4*), but occurs infrequently. Interestingly, the *sna* promoter is also moderately paused in our ChIP assay and at its endogenous locus[74], but this promoter is not regulated by a three promoter-state dynamic. We hypothesize that pausing is highly unstable for this promoter at this stage and does not constitute a rate-limiting step that we can capture with our live-imaging assay (see "Discussion").

In addition to two separable inactive states, the *kr* transgene exhibited a highly probable permissive state (0.83) with a duration of 94 s (Fig. 6g, Supplementary Fig. 5a, and Supplementary Table 2). When the INR of *kr* was mutated (*kr-INR1*), there was a significant increase in ON duration (Fig. 6j and Supplementary Table 2). A similar effect was obtained with an alternative mutation of the INR (*kr-INR2*) (Supplementary Figs. 5a and 6m and Supplementary Table 2).

Interestingly, in all the transgenes, the differences in the number of states and the active/inactive durations were not accompanied by a change in Pol II firing rates. The *sna*, *sna+INR*, *kr*, and *kr-INR1* mutant all exhibited an estimated $k_{INI}$ of 1 Pol II every 8–9 s (Fig. 6f–g, i–j and Supplementary Table 2). These estimates agree with recently documented polymerase initiation rates of *Drosophila* developmental promoters[30,32,75]. Our results suggest the rate of initiation is not associated with the presence or absence of an INR motif. Instead, the INR motif leads to a regulation of transcriptional bursting via two inactive promoter states, one of which is associated with stabilized pausing.

Taken collectively, our results demonstrate that transcription dynamics are differentially regulated by the INR and the TATA box. Developmental promoters containing a strong INR motif can travel through multiple inactive states due to an extra regulatory step at early elongation via Pol II pausing. Our live-imaging data favor a model whereby pausing occurs on the order of minutes but is not an obligatory state reached by all engaged polymerases. Instead, we propose that when pausing constitutes a rate-limiting step, it occurs for only a subset of polymerases with a low frequency.

**Promoter sequences and timing of initiation.** A second property that we found to be common to all promoters examined, was the regime of waiting times between mitosis and first transcriptional activation. The analysis of the lag time between mitosis and onset of activation is "non-stationary" and was previously modeled using a different modeling paradigm, based on mixed gamma distributions of the random time to transcription activation after mitosis in nc14[57]. This analysis estimates two parameters, the number of promoter states (a) and transition times between them (b). We applied this

modeling framework to our data by only focusing on the distribution of waiting times between mitosis and first activation (illustrated in Supplementary Fig. 10a). Interestingly, a multiexponential distribution becomes a mixed gamma distribution when the time parameters of the exponentials are even. Conversely, the equality of these time parameters justifies the use of a mixed gamma regression. Using the more general multiexponential regression, we proved that regardless of the promoter sequence, the transition times between states during this lag regime are homogeneously distributed (Supplementary Fig. 10b, c). This is in contrast with the transition times between states in the stationary bursting regime, which are heterogeneous. This suggests that the states and transitions involved in the two regimes (nonstationary and stationary) are distinct and have separate regulations. This further supports a previous study that demonstrated the delay in postmitotic transcription activation is dependent on the enhancer sequence[57].

**Interplay between TATA and INR**. Recent single-cell RNA-seq quantification suggests that promoters exhibiting both TATA and INR motifs produce higher burst sizes than when only one motif is present[67]. However, systematic interrogation of human promoters with a synthetic biology approach shows that TATA and INR additively but not synergistically increase gene expression[76]. A synergistic and/or additive effect does not seem to be evident in *Drosophila*, as TATA-containing promoters have relatively infrequent INR motifs or exhibit an INR sequence devoid of a G at +2, shown to be associated with long pause durations[12,50].

To quantify TATA box and INR interplay on rate-limiting promoter switching states in vivo, we compared the *sna + INR* and *snaTATAlight* transgenes to that of a combined mutant *snaTATAlight+INR* promoter (Supplementary Fig. 7a). Remarkably, after applying the deconvolution procedure to *snaTATAlight +INR* nuclei (Supplementary Fig. 7g), the distribution of waiting times between initiation events required a fit with two exponentials similarly to *snaTATAlight* (Supplementary Fig. 7k–k'). We note however that the statistical tests allocating the number of promoter states for this particular genotype were not robust (Supplementary Table 1). In terms of promoter kinetics, the *snaTATAlight+INR* behaved similarly to the *snaTATAlight* promoter (Supplementary Fig. 5a, b). Interestingly, the *snaTATAlight+INR* has a p(OFF) nearly twice that of the *snaTATAlight* promoter (Supplementary Table 2), indicating that the INR may act to maintain a nonpermissive promoter state even in the absence of polymerase pausing.

To gain more insight into TATA/INR interplay, we also generated a *kr* mutant promoter that does not contain a non-canonical TATA (*kr-TATA*; Supplementary Fig. 6a). Data from this promoter required a three-state model (Supplementary Fig. 6k–k'), similar to the *kr* promoter. In the *kr-TATA* promoter, the duration and probability of the paused state increased while the ON state appeared shorter and less probable (Supplementary Fig. 5a, c and Supplementary Table 2). This indicates that the loss of TBP binding may increase pausing induced by the INR motif, potentially providing a mechanistic rationale for the relative scarcity of dual TATA/canonical INR promoters[12,50] in the *Drosophila* genome. Taken together, these results indicate that TATA and INR are not simply synergistic or additive but display more complex interactions.

## Discussion

The spatiotemporal organization of gene expression is critical to the development of a functional living organism. While we have accumulated knowledge on how enhancers precisely regulate gene expression, we know relatively little about the impact of core promoters on transcriptional bursting in a developing embryo. Here, we investigated how minimal 100 bp sequences of developmental promoters control transcriptional states and their switching kinetics. We found that a classical two-state model does not suffice to capture promoter dynamics of stably paused promoters containing a strong INR motif but does fit with TATA-containing promoters.

The development of a novel numerical deconvolution method revealed startling insights into the variation of transcriptional initiation between minimal promoter sequences in vivo. Our experimental setup allowed us to unmask gene expression variations, which are only dictated by variations in core promoter sequences. Indeed, cell cycle duration and synchrony, the concentration of input transcription factors, and chromatin states were intentionally kept constant. Our assay revealed that a similar transcriptional activity profile could be obtained from two very distinct promoter sequences via distinct modes of transcriptional initiation. While *sna* promoter dynamics could be explained by a simple two-state model, the *kr* promoter required a three-state model, with two distinct inactive promoter states, a short-lived and a longer-lived one. These distinct transcriptional regimes are unlikely due to differences in enhancer–promoter specificity, as evidenced by the synchronous and high transcriptional activation reached with these two promoters. However, knowing the widespread preferential promoter code for specific enhancers[77,78], it will be interesting to use our pipeline to examine which rate-limiting step of promoter dynamics is tuned by enhancer–promoter choice.

How could such a small number of states be compatible with the numerous promoter-binding events occurring during transcription initiation? Biochemical studies reveal structural states, while imaging-based approaches unmask key rate-limiting kinetic states. Whole-genome methylation footprinting disentangled five transcription initiation states in *Drosophila* cultured cells[15]. Remarkably, the authors demonstrated that TATA-containing promoters were frequently found in PIC-bound configuration (PIC alone or PIC + Pol II). We, therefore, propose that the permissive state in *Drosophila* embryos corresponds to a state where promoter DNA is bound by the PIC. The inactive state, which we found to be very transient, could then possibly represent a TBP-unbound configuration consistent with TBP dynamics observed in human cells[37,39,79].

By using the *sna* promoter as a model, we found that TATA box directly impacts promoter occupancy of the active state and allows large transcriptional bursts by promoting long ON durations. How could the presence of a TATA box permit these long ON durations? Slow TBP protein turnover at human TATA box-containing genes may foster stable long ON durations[37,39,79]. However, this hypothesis needs to be confirmed by quantification of TBP kinetics during *Drosophila* ZGA.

In its endogenous context, the *snail* promoter is among the first genes to be transcribed during ZGA, in a particularly constrained environment with extremely short cell cycles (<15 min). Remarkably, the majority of genes expressed during this critical period are highly expressed, short, and intron-less with a canonical TATA box and are generally nonpaused[18,50]. Subsets of these, including *snail*, are considered "dual promoters", as they gradually acquire pausing as development proceeds[50]. Thus, it is possible that the kinetic bottlenecks regulating transcription of developmental promoters evolve as developmental timing proceeds, with paused polymerase gradually emerging as a rate-limiting step. In our study, the promoter model *sna* shows a moderate level of pausing (as measured by ChIP) but is regulated by a simple two-state model in nc14. This could indicate that pausing occurs on a timescale indistinguishable from the OFF state and is thus embedded within it. Strikingly, early transcription in zebrafish embryos occurs in similarly constrained rapid cell cycles where a subset of zygotically activated genes also

display a T/A-rich WW-box motif[80]. Thus, regulation of transcription via a unique rate-limiting step (OFF to ON transition) that is TBP-dependent might be conserved between fly and vertebrate embryos.

In this work, we provided quantitative evidence that INR-containing promoters are associated with a three-state model (ON, paused, OFF), contrasting with TATA promoters.

Interestingly, genomics studies revealed that transcription initiation code evolves during ZGA in flies and vertebrates[50,54,80]. Given the results of this study and those obtained in *Drosophila* cultured cells[12,16], it is tempting to link the gradual emergence and stabilization of pausing during ZGA with the presence of an INR, in particular an INR with a G at +2 position[12]. In light of our results, we propose that the switch in promoter usage from TATA-driven to INR-driven during ZGA may lead to a change in transcriptional dynamics from two to three states, to include an elongation-mediated checkpoint. Acquisition of an extra rate-limiting step might help control cell-to-cell expression variability as well as fine-tune gene expression levels.

Taken collectively, this study establishes our promoter imaging assay and novel mathematical deconvolution and modeling methodology as a valuable tool to probe gene expression dynamics during development. Quantitative analysis of promoter dynamics at high temporal resolution opens the door to a deeper insight into the molecular mechanisms underlying transcriptional regulation in vivo. Future studies involving direct manipulation of pause initiation or duration in living embryos using approaches such as optogenetics[81] combined with this framework will help to establish a broader understanding of the nature of promoter states and the role of Pol II pausing in vivo.

## Methods

**Drosophila stocks and genetics**. All crosses were maintained at 25 °C. Transgenic lines were maintained as homozygous stocks. For live imaging, homozygous males carrying the transgene of interest were crossed with homozygous females bearing the *MCP-eGFP-His2Av-RFP* construct. For smiFISH, homozygous flies were crossed to *yw* in order to facilitate single-molecule detection. *nos*:GAL4-VP16 was recombined with *MCP-eGFP-His2Av-RFP*[57]. UAS:*white RNAi* (#35573) and UAS:*Nelf-A RNAi* (#32897) were obtained from the Bloomington Stock *Drosophila* Centre (University of Indiana, Bloomington).

**Cloning and transgenesis**. The *sna* distal enhancer-24xMS2-*y* minigene has been previously described[51,57]. Promoters (Supplementary Fig. 1 and Supplementary Table 3) were amplified from genomic DNA using Q5 polymerase (New England Biolabs) and inserted between the enhancer and 24xMS2 sequences using restriction enzyme-mediated ligation. Mutations were performed with the QuikChange II Site-Directed Mutagenesis kit (Agilent Technologies) or synthesized (Twist Biosciences) and inserted using restriction-mediated cloning. All constructs were sequenced to ensure appropriate insertion. Transgenic flies were generated using PhiC31-mediated recombination (Best Gene, Inc.), and all constructs were inserted into the same genetic background and genomic position (*BL 9750*). Stocks are available upon request.

**Live imaging**. Embryos were permitted to lay for 2 h prior to collection for live imaging. Embryos were hand dechorionated and mounted on a hydrophobic membrane prior to immersion in oil to prevent desiccation, followed by the addition of a coverslip.

Live imaging was performed with an LSM 880 with Airyscan module (Zeiss). Z-stacks comprised of 30 planes with a spacing of 0.5 µm were acquired at a time resolution of 3.86 s per stack in fast Airyscan mode with laser power measured using a ThorLabs PM100 optical power meter (ThorLabs Inc.), and maintained across embryos at 3.8 µW for constitutive MCP-GFP/His2A-RFP expression, and 5.0 µW for RNAi analyses. All movies were performed with the following settings: GFP excitation by a 488-nm laser and RFP excitation by a 561 nm were captured on a GaAsP-PMT array with an Airyscan detector using a ×40 Plan Apo oil lens (NA = 1.3) and a 3.0 zoom on the ventral region of the embryo centered on the presumptive ventral midline. Resolution was 512 × 512 pixels with bidirectional scanning. Airyscan processing was performed using 3D Zen Black v3.2 (Zeiss).

**Live-imaging analysis**. Visualization and analysis of the time series were performed using a custom-made software developed in Python™ that permits visualization of each analysis step and manual correction if necessary

(Supplementary Fig. 2). Activation time traces were collected starting from the Airyscan-processed Z-series described above. Green (MS2) and red (His2Av) channels were clipped to consider only after the start of nc14 as defined by the progression of anaphase across the region of interest. Nuclei were maximum intensity projected and pre-smoothed with a Gaussian filter and then thresholded with an Otsu threshold value. The resulting connected components of the binary images were then labeled and touching nuclei segmented with a watershed algorithm. The software enabled manual correction of the segmentation. Nuclei were finally tracked across the time frames using a minimum distance criterion plus a user-defined distance threshold. Nuclei appearing for only a few frames or those touching the border were removed from the analysis to exclude sources of errors.

GFP puncta representing transcription sites were analyzed in 3D. Because the duplication of DNA occurs relatively early in nc14 and because of the immediate proximity of sister chromatids[30,82], it is challenging to independently resolve individual sister chromatid signals using live imaging. This is why the GFP signal at each transcriptional spot can be considered as the sum of both sister chromatids that we treat as a single transcriptional trace. For each time frame, the 3D image was filtered with a 3D Laplacian of Gaussian filter and then thresholded. The threshold value (THR) was expressed as $\mu + \text{THR} * \sigma$, where $\mu$ and $\sigma$ are the average and the standard deviation of the pixels values of the filtered image respectively, while THR is a user-defined value. The threshold value was in this way rescaled with respect to statistical properties of the filtered image, making THR a value independent of the particular data acquisition. All detected spots were filtered to remove (1) all the spots with a volume less than a user-defined volume threshold, and (2) spots present in only one z-frame. For each time frame, detected spots were associated in 2D to the overlapping or closest nucleus, inheriting the tracking from them; a user-defined distance threshold between nucleus and spot was used in order to avoid mis-associations.

Finally, each nuclei-puncta pair passing filtering was described as a time series of intensity, volume, and position. To eliminate intensity variation within the Z-series, spot intensity values were divided by the background fluorescence of the average intensity value of the pixels surrounding the independent spots. Analysis was restricted to the region 25 µm on either side of the center of the gastrulation furrow present at the end of nc14 as positioned using a maximum intensity tile-scan of the entire embryo to determine the coordinate position of the Z-stack. From this data, it is possible to extract: the timing of activation measured as the first timepoint GFP fluorescence crosses the software detection threshold for >1 Z-stack; cumulative activation; intensity profiles for individual nuclei; and individual nuclei burst trajectories. Integral amplitude was calculated using R to determine the surface area under the curve of each individual nuclei and analyzed with Prism (Graphpad 8.0.1) using a Kruskal–Wallis test for significance with multiple comparison adjustments. The calibration method has a minimum detection threshold of >3 transcripts per transcription site. All figures report the number of nuclei and movies used for analysis in figure legends. Movies of genotypes not supplied as Supplementary Movies available upon request.

**smiFISH**. Embryos heterozygous for the transgene of interest were fixed in 10% formaldehyde/heptane for 25 min with shaking before a methanol quench and stored at −20 °C in methanol before use. smiFISH probes targeting the *yellow* (*y*) reporter gene were designed using Oligostan[83–85] with FLAP-Y for secondary probe recognition (Integrated DNA Technologies, Inc.). Probe sequences are provided in Supplementary Table 4. Secondary probes were conjugated to Cy3 at the 5' and 3' ends (Integrated DNA Technologies, Inc.). Probes were resuspended in TE at appropriate equimolar concentrations. Prior to probe addition to embryos, the *y* targeting primary probes were hybridized to the secondary FLAP-Y probes as described[83] and maintained at −20 °C in the dark prior to use.

Embryos were prepared for smiFISH as briefly follows: embryos were dehydrated with 2 × 5 min washes in 100% ethanol, followed by rehydration in PBT for 4 × 15 min and equilibration in 15% formamide/1 × SSC for 15 min. During equilibration, the smiFISH probe mixture was prepared with a final concentration of 1 × SSC, 0.34 µg µL⁻¹ *E. coli* tRNA (New England Biolabs), 15% formamide (Sigma), 5-µL probe duplex, 0.2 µg µL⁻¹ RNAse-free BSA, 2 mM vanadyl-ribonucleoside complex (New England Biolabs), and 10.6% dextran sulfate (Sigma) in RNAse-free water. The equilibration mixture was removed and replaced with probe mixture, and embryos were incubated overnight in the dark at 37 °C. The following day, embryos were rinsed twice in equilibration mix and twice in PBT, followed by DAPI staining and three PBT washes before mounting in ProLong Gold mounting media (Life Technologies).

**Fixed sample imaging and analysis**. Fixed sample imaging was performed on an LSM 880 with Airyscan module (Zeiss). Z-planes were acquired with 0.20 µm spacing to a typical depth of 80–100 Z-planes from the apical surface of the embryo, using laser scanning confocal in Airyscan super-resolution mode with a zoom of 3.0. DAPI excitation was performed with a 405-nm laser and Cy3 excitation with a 561-nm laser, with detection on a GaAsP-PMT array coupled to an Airyscan detector. Airyscan processing was performed using 3D Zen Black v3.2 (Zeiss) prior to analysis. Embryos were staged based on membrane invagination. Z-stacks were taken at both the center of the presumptive mesoderm as well as at the border region.

Images were analyzed (Supplementary Fig. 3) with Imaris v9.2.2 by first determining the threshold of detection using the non-mesodermal border region.

After applying the threshold to the center pattern, fixed-sized shells (XY radius >0.3 μm, Z radius of >1.0 μm) were created around the centroid of detected objects. The median signal intensity of object shells was used as a proxy for the intensity of single molecules of RNA. The transcription site intensity of each nucleus was summed to account for the presence of sister chromatids and treated as a single transcription site throughout. The mean transcription site intensity was divided by the median single-molecule intensity to determine the average number of mRNA molecules present at the transcription sites.

### Mathematical modeling of burst parameters and multiexponential regression fitting

*Burst deconvolution and pol II positioning.* The Pol II positions were found by combining a genetic algorithm with a local optimization procedure[59].

Before the initiation of the analysis algorithm, several key parameters were established. The Pol II elongation speed was fixed at 45 bp s$^{-1}$(see ref. [60]). The reporter construct transcript was divided into three sections consisting of the pre-MS2 fragment (41 bp), 24xMS2 loops (1292 bp), and post-MS2 fragment containing the *yellow* reporter (4526 bp). The retention time was assumed to be small in relation to the time needed to produce a transcript, and so was fixed at 0 s. The temporal resolution of each movie was 3.86 s per frame. This frame rate is sufficient to detect processes that occur on the order of seconds.

The possible polymerase positions were discretized using a step of 30 bp (or equivalently 2/3 s). This step was chosen, as it is smaller than the minimum polymerase spacing and large enough to have a reasonable computation time. For a movie of 35-min length, this choice corresponds to a maximum number of 3150 positions.

The algorithm was implemented in Matlab R2020a using Global Optimization and Parallel Computing Toolboxes for optimizing Pol II positions in parallel for all nuclei in a collection of movies. The resulting positions are stored for analysis in the further steps of our computational pipeline. At this step, the density of Pol II initiation events can be visualized by binning time and checking the occurrence of Pol II activation in each bin. This was rendered as a heatmap in which rows represent a single-nucleus time series and the number of activation events per 30-s bin (or equivalently 1350 bp) is indicated by the color (Fig. 2e).

The deconvolution step is common to all of the MS2 data analysis pipelines. A detailed description of the algorithm can be found in ref. [59].

*Multiexponential regression fitting of the survival function and model reverse engineering using the survival function.* Data from several movies corresponding to the same genotype was first pooled together. The entry and exit of each trace corresponding to a unique nucleus were defined using a threshold representing 1/5 of the maximum intensity for the specific trace, in order to restrict the analysis to the stable part of the signal (Fig. 2a, gray box). Waiting times were extracted as differences between successive Pol II positions from all the resulting traces and the corresponding data was used to estimate the nonparametric cumulative distribution function by the Meyer–Kaplan method. This also permits the calculation of a 95% confidence interval for the experimental survival function that is further used to judge the quality of a parametric multiexponential regression fitting.

Then, a multiexponential regression fitting produced a set of $2N - 1$ distribution parameters, where $N$ is the number of exponentials in the regression procedure (3 for $N = 2$ and 5 for $N = 3$). The regression procedure was initiated with multiple initial guesses and followed by local gradient optimization of the following objective function[59]:

$$O = \frac{\alpha}{n}\sum_{i=1}^{n}(S(t_i) - S_e(t_i))^2 + \frac{1-\alpha}{n}\sum_{i=1}^{n}(\log(S(t_i)) - \log(S_e(t_i)))^2, \quad (1)$$

where $S(t_i), S_e(t_i)$ are the theoretical (multiexponential) and empirical (estimated by the Meyer–Kaplan method) survival functions, respectively, and $\alpha$ is a parameter satisfying $0 \leq \alpha \leq 1$ and representing the weight of linear scale differences in the objective function. We chose an intermediate value $\alpha = 0.6$ for all our parameter estimates (these estimates are nevertheless robust with respect to $\alpha$).

The optimization resulted in a best-fit solution with additional suboptimal solutions (local optima with objective function value larger than the best fit). A multiexponential regression is considered acceptable if the predicted survival function is within the confidence bounds of the experimental survival function. This provides a method to select the number $N$ of exponentials in the regression: we progressively increase $N$ starting with $N = 2$ until an acceptable regression is reached.

The $2N - 1$ distribution parameters can be computed from the $2N - 1$ kinetic parameters of a $N$-state transcriptional bursting model. Conversely, a symbolic solution for the inverse problem was obtained, allowing computation of the kinetic parameters from the distribution parameters and reverse engineering of the transcriptional bursting model. In particular, it is possible to know exactly when the inverse problem is well-posed, i.e., there is a unique solution in terms of kinetic parameters for any given distribution parameters in a domain.

The transcriptional bursting models used in this paper are as following (see Tantale et al.[59] for a detailed description):

For $N = 2$, there were three distribution parameters and three kinetic parameters.

The distribution parameters are $A_1, \lambda_1, \lambda_2$, defining the survival function

$$S(t) = A_1 e^{\lambda_1 t} + (1 - A_1)e^{\lambda_2 t}. \quad (2)$$

The solution of the inverse problem for the ON–OFF telegraph model (Fig. 2k and Fig. 6b) is

$$k_2 = -S_1, \quad (3)$$

$$k_1^- = S_1 - \frac{S_2}{S_1}, \quad (4)$$

$$k_1^+ = \frac{S_3 S_1 - S_2^2}{S_1(S_1^2 - S_2)}, \quad (5)$$

$$S_1 = A_1\lambda_1 + A_2\lambda_2, \quad (6)$$

$$S_2 = A_1\lambda_1^2 + A_2\lambda_2^2, \quad (7)$$

$$S_3 = A_1\lambda_1^3 + A_2\lambda_2^3, \quad (8)$$

$$A_2 = 1 - A_1, \quad (9)$$

where $k_2, k_1^+, k_1^-$ are the initiation rate, the OFF to ON and ON to OFF transition rates, respectively. Thus, the duration of the OFF and ON states can be calculated as:

$$T(\text{OFF}) = \frac{1}{k_{1+}}, \quad (10)$$

$$T(\text{ON}) = \frac{1}{k_{1-}}. \quad (11)$$

For this model, the probability to be in the state ON and OFF is:

$$p_{\text{ON}} = \frac{k_1^+}{k_1^+ + k_1^-}, \quad (12)$$

$$p_{\text{OFF}} = 1 - p_{\text{ON}}. \quad (13)$$

For $N = 3$, there were five distribution parameters and five kinetic parameters[59].

The distribution parameters are $A_1, A_2, \lambda_1, \lambda_2, \lambda_3$, defining the survival function

$$S(t) = A_1 e^{\lambda_1 t} + A_2 e^{\lambda_2 t} + (1 - A_1 - A_2)e^{\lambda_3 t}, \quad (14)$$

The inverse problem has a unique solution for the three-state model (non-obligatory pause) with two OFF states (OFF and PAUSE) and one ON state (Fig. 6c)

$$k_3 = -S_1, k_2^+ = \frac{1}{2}\left[-L_1 + \frac{S_2}{S_1} - \frac{\sqrt{(S_1 L_1 - S_2)^2 - 4L_3 S_1}}{S_1}\right], \quad (15)$$

$$k_2^- = \frac{1}{2}\left[S_1 - \frac{S_2}{S_1} + \frac{-S_1^2 L_1 + S_1 S_2 + S_1 L_2 - L_3 + \frac{S_2^2}{S_1} - S_3}{\sqrt{(S_1 L_1 - S_2)^2 - 4L_3 S_1}}\right], \quad (16)$$

$$k_1^+ = \frac{1}{2}\left[-L_1 + \frac{S_2}{S_1} + \frac{\sqrt{(S_1 L_1 - S_2)^2 - 4L_3 S_1}}{S_1}\right], \quad (17)$$

$$k_1^- = \frac{1}{2}\left[S_1 - \frac{S_2}{S_1} - \frac{-S_1^2 L_1 + S_1 S_2 + S_1 L_2 - L_3 + \frac{S_2^2}{S_1} - S_3}{\sqrt{(S_1 L_1 - S_2)^2 - 4L_3 S_1}}\right], \quad (18)$$

where

$$S_1 = A_1\lambda_1 + A_2\lambda_2 + A_3\lambda_3, \quad (19)$$

$$S_2 = A_1\lambda_1^2 + A_2\lambda_2^2 + A_3\lambda_3^2, \quad (20)$$

$$S_3 = A_1\lambda_1^3 + A_2\lambda_2^3 + A_3\lambda_3^3, \quad (21)$$

$$A_3 = 1 - A_1 - A_2, \quad (22)$$

$$L_1 = \lambda_1 + \lambda_2 + \lambda_3, \quad (23)$$

$$L_2 = \lambda_1^2 + \lambda_2^2 + \lambda_3^2, \quad (24)$$

$$L_3 = \lambda_1^3 + \lambda_2^3 + \lambda_3^3. \quad (25)$$

and $k_3, k_2^+, k_2^-, k_1^+, k_1^-$ are the transcription initiation, PAUSE to ON, ON to PAUSE, OFF to ON, and ON to OFF rates, respectively.

Thus, duration of the ON, OFF, and PAUSE states can be calculated as:

$$T(\text{OFF}) = \frac{1}{k_{1+}}, \tag{26}$$

$$T(\text{PAUSE}) = \frac{1}{k_{2+}}, \tag{27}$$

$$T(\text{ON}) = \frac{1}{k_{1-} + k_{2-}}. \tag{28}$$

For this model, the steady-state probability to be in a given promoter state is

$$p_{\text{OFF}} = \frac{k_1^- k_2^+}{k_1^+ k_2^+ + k_1^- k_2^+ + k_1^+ k_2^-}, \tag{29}$$

$$p_{\text{PAUSE}} = \frac{k_1^+ k_2^-}{k_1^+ k_2^+ + k_1^- k_2^+ + k_1^+ k_2^-}, \tag{30}$$

$$p_{\text{ON}} = \frac{k_1^+ k_2^+}{k_1^+ k_2^+ + k_1^- k_2^+ + k_1^+ k_2^-}. \tag{31}$$

The alternative three-state model with systematic pause (Supplementary Fig. 5) satisfies the following relation among distribution parameters[59]:

$$A_1\lambda_1 + A_2\lambda_2 + (1 - A_1 - A_2)\lambda_3 = 0. \tag{32}$$

This means that only four and not five distribution parameters are free, which further constrains the three-exponential fittings. In order to infer this model, a constrained fitting was performed but the bad quality of fitting recommended rejection of the model (Supplementary Fig. 5).

*Testing the method with artificial data.* The entire computational pipeline was tested using artificial data (see also Tantale et al.[59]). Artificial traces were generated by simulating the model using the Gillespie algorithm with parameter sets similar to those identified from data. The simulations generated artificial polymerase positions, from which a first version of the signal was computed by convolution. The results are provided in Fig. 2e–h.

*A modified Kolmogorov–Smirnov test for the parametric distribution.* A one-sided one-sample Kolmogorov–Smirnov (KS) test was used to check if the waiting times follow the parametric fitted distribution. The output of this test is a *P* value that is large if the waiting times follow the fitted distribution, and small if not. The KS test is based on differences between estimated and empirical probabilities, being thus sensible to errors in the larger probabilities of shorter waiting times, and less sensitive to rare, very long waiting times. Our fitting procedure combines linear and logarithmic scales to find a balance between short and long time scales (see Tantale et al.[59]). Although relatively small (in terms of the objective function), the resulting errors on short times are large enough to be considered significant by the KS test for all models. In order to be able to distinguish between models, we have limited the analysis to times larger than 10–20 s.

*Error intervals.* Distribution parameters result from multiexponential regression fitting using gradient methods with multiple initial data. These optimization methods provide the best fit (global optimum) but also suboptimal parameter values. Using an overflow ratio (a number larger than one, in our case 2) to restrict the number of suboptimal solutions, we define boundaries of the error interval as the minimum and maximum parameter value compatible with an objective function less than the best-fit times the overflow.

**Mathematical modeling of postmitotic gaps.** The distribution of the postmitotic gap was estimated using the equation below. The fitting suggests that the timescale parameters of the postmitotic gap are even, i.e., $\lambda_1 = \lambda_2 = \lambda_3$. In this limit, the five parameters, three-exponential distribution defined by the equation below, becomes the simpler, three parameters mixed gamma distribution described as

$$S(t) = p_1 e^{-\frac{t}{b}} + p_2\left(1 - \gamma\left(2, \frac{t}{b}\right)\right) + (1 - p_1 - p_2)\left(1 - \frac{\gamma\left(3, \frac{t}{b}\right)}{2}\right), \tag{33}$$

where

$$\gamma(a, x) = \int_0^x s^{a-1} e^{-s} ds \tag{34}$$

is the lower incomplete gamma function, and $p_1, p_2, b$, are the probabilities of one step, two steps, and the mean step duration, respectively.

**Chromatin immunoprecipitation.** Homozygous embryos were collected and fixed in 1.4% formaldehyde for 20 min prior to dry storage at −80 °C. Embryos were dissociated on ice in RIPA buffer (150 mM NaCl, 1.0% IGEPAL CA-630, 0.5% sodium deoxycholate, 0.1% SDS, 50 mM Tris-HCl pH 7.8) supplemented with

protease and phosphatase inhibitors (Roche). Chromatin shearing was performed in a pre-chilled Bioruptor Pico (Diagenode) for five cycles of 30 s/30 s. Samples were divided into equal volumes and incubated overnight at 4 °C with either rabbit anti-Rbp3 (10 μg) or rabbit anti-Nelf-E (10 μg) (gifts from J. Zeitlinger[16]) or normal serum. A second incubation overnight at 4 °C with Protein G-magnetic Sepharose beads (GE Healthcare) was used to pull down protein:DNA complexes prior to washing in low salt (120 mM NaCl, 0.1% SDS, 0.5% Triton X-100, 20 mM Tris-HCl pH 8.0) and high salt (500 mM NaCl, 0.1% SDS, 0.8% Triton X-100, 20 mM Tris-HCl pH 8.0) buffers, elution, incubation with protease K and RNAse A, and DNA retrieval using the QiaQuick PCR cleanup kit (Qiagen). qPCR analysis was performed using Light Cycler 480 SYBR Green I Master Mix (Roche) using primers listed in Supplementary Table 5. Analysis was performed using Microsoft Excel and Prism (Graphpad v8.0.1) with a Mann–Whitney test to determine significance.

**Quantitative RT-PCR.** To test RNAi-mediated knockdown of Nelf-A expression, 0–2 h embryos were homogenized in Trizol (Invitrogen) and RNA was extracted per the manufacturer's directions. Reverse transcription was performed using the Superscript IV system (Invitrogen) with oligo d(T)$_{20}$ priming prior to qPCR analysis. *nos:*GAL4-VP16; UAS:*white* RNAi embryos were used as the control and all measurements were performed in biological and technical triplicate. qPCR analysis was performed using Light Cycler 480 SYBR Green I Master Mix (Roche) using primers listed in Supplementary Table 5. Analysis was performed using Microsoft Excel and Prism (Graphpad 8.0.1) with a one-tailed Student's *t* test to determine significance.

**Image analysis software.** Live imaging analysis software is available at http://www.igmm.cnrs.fr/segment-track/ along with a video tutorial. Code for the mathematical analysis of burst parameters and multiexponential regression fitting is available[59].

**Reporting summary.** Further information on research design is available in the Nature Research Reporting Summary linked to this article.

## Data availability
The data that support this study is available from the corresponding author upon reasonable request. Source data are provided with this paper.

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

## Acknowledgements

We thank Julia Zeitlinger, Jean-Christophe Andrau, Jeremy Dufourt, and all members of the Lagha lab for their critical reading of the manuscript and constructive discussions. We are grateful to F. Juge for the insightful discussion—and H. Faure-Gautron for technical assistance. We acknowledge the MRI imaging facility, a member of the national infrastructure France-BioImaging supported by the French National Research Agency (ANR-10-INBS-04, «Investments for the future»). This work was supported by the ERC SyncDev starting grant to M.L and CNRS; by CNRS grant PEPS MIGHTY to O.R. and E. B., and by ANRS grants ECTZ62561 to E.B.. O.R. acknowledges support from the French National Research Agency (ANR-17-CE40-0036, project SYMBIONT). Machine-learning calculations were performed on the HPC facility MESO@LR run by the University of Montpellier.

## Author contributions

Conception and design: M.L.; acquisition of the data: M.D., V.P., and C.F.; analysis: M.L., M.D., V.P., and O.R.; software: A.T. and O.R.; modeling: O.R.; interpretation of the data: M.L., M.D., V.P., and O.R.; writing: V.P. and M.L. with input from O.R. and E.B.; visualization: M.D., V.P., and O.R.; supervision: M.L.; fund acquisition: M.L. O.R. and E.B. conceived and developed the machine-learning deconvolution procedure and the facultative model of paused polymerase. All authors read the manuscript.

## Competing interests

The authors declare no competing interests.
