## [Peer Review File · Nature Communications]

Reviewers' Comments:

Reviewer #1:

Remarks to the Author:

In this manuscript, Dejean and colleagues perform live imaging of transcription in drosophila embryos to dissect the relationship between promoter structure and transcriptional dynamics. They focus on the role of the TATA box and the INR elements. Using mathematical modelling to infer the changes in RNA Pol-II occupancy on their reporter genes, they found that transcription driven by the TATA box can be modelled by a two-state transcription model. They suggest that the presence of INR is linked to a higher propensity for Pol II pausing and leading to a three-state model of transcription. A state of pausing for a fraction of polymerases is also proposed. Overall the manuscript is well-written, and the imaging experiments and analysis are of high quality. The main novelty of this manuscript lies in the connection between promoter structure and transcriptional bursting dynamics in the context of drosophila development. While the presence of an INR is convincingly linked to the three-state model, how the INR leads to this additional state is not clear (see first major point below).

Major points:

- While the conclusions regarding the link between the presence of an INR and a third state of promoter activity is convincing, the part of the manuscript addressing the pausing itself is less so. Fig.6C' shows that the presence of an INR increases the pausing index, however, this pausing index is the same for the sna+INR and the kr-INR1 constructs. The sna+INR is better fitted by a three-state model, and the kr-INR1 by a two state model. Thus the pausing itself does not seem correlated to the three state model, which seems at odds with their model in Fig.6B'.
- There is genome-wide data on pausing in drosophila - this data should be cited and/or reanalysed to shed light on the role of the INR in pausing.
- The data to back up the pausing analysis is too light. ChIP-QPCR data presented in Fig.6C' should show all data points, and more detailed data should be shown in the supplement (genome tracks, quantification of Pol II density). The authors used a t-test for this data; this is not appropriate since the normality of the distributions cannot be assessed with only 3 replicates.
- I appreciate that the authors performed calibration of their MS2 signal on RNA FISH, but there is no mention of the actual sensitivity of their live MS2 measurements. What is the minimal number of mRNAs at the transcription spot they require to see a fluorescent spot ? If that threshold is $\gg 1$, how does this affect their inferences of on-off switching, in particular for the strains with very low transcriptional activity ?

Other points:

- What are the error ranges for Fig.4J-L, Fig. 6D and 6J ? Some statistics are needed here
- In the text of Fig.1F, the authors mention that brk goes up to 17% of activation at the end of nc14. However from the plots it seems to reach about 25%.
- In Supplemental Figure 1, the x-axis for the CAGE data should be labelled, as well as the y-axis of the TBP ChIP-seq data.
- I suggest avoiding subjective wording such as "incredibly" (second half of page 10).
- I suggest to briefly mention what the yellow reporter gene is - will be helpful for non-drosophilists

Reviewer #2:

Remarks to the Author:

Dejean et al implemented the MS2/MCP live imaging system and a machine-learning based Markovian promoter state transition model of transcription to characterize how promoter composition dictates transcriptional dynamics in *Drosophila* embryos. They studied two promoter motifs – the TATA box and the INR. The reporter gene driven by the same snail minimal enhancer produces distinct transcriptional activities upon various promoter compositions. TATA box-containing promoters (e.g. snail promoter) exhibit long active states with high Pol II firing rates and short inactive states. Transcription from the TATA box-containing promoters can be described with a simple two-state model. On the other hand, the INR-containing promoters demonstrate three-states, with two inactive and one active state. They suggested that the second inactive state is associated with the promoter-proximal pausing, exhibiting inactive period of a few minutes. Lastly, they showed that not all polymerases pause, but only a subset of them enters a paused state.

The authors present interesting and noteworthy results that investigate the role of different promoter motifs in transcriptional regulation. This highly quantitative work is significant to the field of transcriptional regulation and the use of mathematical modeling helps figure out the implication of their experimental observations. Yet, I believe that some of their findings do not necessarily support their interpretations and conclusions. For example, while the INR motif supposedly increases the pausing, more transcripts were produced when INR motif was added to the snail promoter. More major comments are mentioned below. Overall, I believe that this work is significant to the field, but the manuscript will need a revision and some clarifications before being published in *Nature Communications*.

Major Comments:

1) According to the previous work from Dr. Lagha, paused promoters facilitate synchronous transcriptional activity. In this work, however, the authors seem to say otherwise, where pausing disrupts the synchronous activity (as seen in delayed activation in INR-containing *kr* and *ilp4* promoters). Or as shown in Figure 5, pausing-associated INR motif does not affect the synchrony. Can the authors clarify this?

2) Similarly, the modeling framework calculates T_{on}/T_{off} and P_{on}/P_{off} after the initiation of transcriptional activity in each nucleus. Then the model does not describe the delay in transcriptional activation seen in different promoters. Do TATA-box and INR motifs regulate the initiation of transcription differentially? Or is the difference due to other parts of the promoter sequence?

3) The authors showed that the promoter with strong TATA mutation showed similar *kini* (Pol II loading rate), and reduced T_{on} . As a result, the burst size is diminished in *snaTATA^{light}* and *snaTATA^{mut}* embryos. Then, with comparable Pol II firing rate, shouldn't the average amplitude be similar among three conditions? In other words, each burst would be longer (hence, larger size), but the amplitude per burst should be similar. This doesn't seem to be the case in Fig 3C and in 4A-C.

4) Based on Figure 5B and C, INR motif does not seem to affect the rate of transcription nor the synchrony, whereas the TATA box motif does change both (Fig 3B and C). If pausing is not responsible for the rate of transcription and the synchrony, what does it do? Does "pausing" simply work to decrease the total mRNA production by having a longer inactive state?

5) In Figure 5C, adding INR to the snail promoter and removing INR from the *kr* promoter both led to increased transcriptional activity. Can the authors explain the observed irony? If INR causes longer inactive state, why is the integral amplitude from *sna*+INR embryos not different from the *sna* embryos?

6) In Figure 6 and in supplemental figures, the difference between the two-state and the three-state

model fit doesn't seem to be that big. In Fig 6A, the two-state model fit is still within the confidence interval. This is also observed in snaTATALight+INR fit (Fig S6K). In fact, fit with the three-state model looks better in all cases including the snail promoter. Is this overfitting? If the three-state model works with the promoters without the INR motif, what do the parameters look like for the "second inactive" state? If the values from the two inactive states are comparable and short, then I think this can be used to show the significance of the "third state" in promoters containing the INR motif.

7) Ton in kr-INR1 embryos is much longer than the Ton from embryos with the snail promoter (Fig 6H). Do you think the non-canonical TATA box in the kr promoter drives stronger transcription (hence longer Ton), but the pausing due to INR-motif dampens it?

Minor Comments:

1) Transcriptional activity from each nucleus is variable, and it's even more so in embryos with promoters that drive low transcriptional activity (e.g., snaTATALight, snaTATAmut) or with INR-motif that has long pausing (e.g., sna+INR or kr). In those cases, the average transcriptional activities shown in Figure 1 or 5 do not represent the bursting (or more precisely, the short/long inactive states). Although the authors demonstrated this with a schematic on promoter states, it'll be nice if some representative traces are shown as well (e.g., I used Figure 4A-C to get better ideas of representative transcriptional activity from each construct).

Reviewer #3:

Remarks to the Author:

This is a very interesting manuscript by Mounia Lagha's group. The literature is rather vague and anecdotal about the roles of core promoter motifs in transcription dynamics- this is in part because a few disparate studies have made conclusions without fully looking at the complexity and redundancy within promoter elements. So, a comparative imaging study, based on minimal promoter fragments in a standardised locus, is very welcome. Another strength of the system is the use of the fly blastoderm, which benefits from the synchrony of measurement and low cellular heterogeneity that could otherwise interfere with clean inferences. In particular, I find the machine learning approach to dealing with MS2 traces very imaginative and innovative. I wish I had thought of doing this.

The data seem very good, and the study is generally clearly written and methodical. The modelling is by and large well explained.

My only substantial concern is that I find the argument linking the 3rd state in the model to a paused state a little weak (bottom of P16 and top of P17). At the moment it seems to be just a fit to a model. Could this not be another inactive promoter state? Can you justify this as a paused state? I can't follow the reasoning here. It might just need tidying up the text for the reader, it might need a better argument/more data. It is hard for me to tell.

Quantitative imaging of transcription in living *Drosophila* embryos reveals the impact of core promoter motifs on promoter state dynamics

Virginia L Pimmett^{1*}, Matthieu Dejean^{1*}, Carola Fernandez¹, Antonio Trullo¹, Edouard Bertrand^{1,3}, Ovidiu Radulescu² and Mounia Lagha^{1#}.

*Equal contribution. #corresponding author/lead contact: mounia.lagha@igmm.cnrs.fr

We are very grateful to the reviewers for their various comments that were extremely helpful to improve our manuscript. We did our best to incorporate the majority of these excellent suggestions. Below we provide a detailed, point-by-point account of the changes in the revised manuscript.

A major point raised by all three reviewers concerned the interpretation of the nature of the third promoter state that we identified. The data supporting our interpretation as pausing were judged to be not totally convincing. To strengthen this important point, we now provide 3 new results to backup this initial interpretation:

- 1-we show the effect of INR mutation in another promoter, *llp4*, along with its pausing status
- 2-we provide new Pol II ChIP-qPCR results, as well as *Nelf-E* ChIP-qPCR on transgenic embryos
- 3-we show the effect of perturbing pausing in *trans*, by using *Nelf-A* RNAi knockdown. This genetic perturbation leads to a change in promoter states for the *kr* promoter, with the disappearance of long waiting times.

Collectively these data reinforce our interpretation of the second inactive promoter state as being associated with pausing.

In addition, we conducted a new statistical analysis (a modified Kolmogorov-Smirnov test) to rigorously determine the number of promoter states (shown in Supplemental Table 1).

Figures have been modified accordingly, with new panels the main figures, 3 new supplemental figures and a new table (Supplemental Table 1). In total, our manuscript is now supported by 10 Supplemental figures, 3 Supplemental tables, and 8 Supplemental movies.

REVIEWER 1

In this manuscript, Dejean and colleagues perform live imaging of transcription in drosophila embryos to dissect the relationship between promoter structure and transcriptional dynamics. They focus on the role of the TATA box and the INR elements. Using mathematical modelling to infer the changes in RNA Pol-II occupancy on their reporter genes, they found that transcription driven by the TATA box can be modelled by a two-state transcription model. They suggest that the presence of INR is linked to a higher propensity for Pol II pausing and leading to a three-state model of transcription. A state of pausing for a fraction of polymerases is also proposed.

Overall the manuscript is well-written, and the imaging experiments and analysis are of high quality. The main novelty of this manuscript lies in the connection between promoter structure and transcriptional bursting dynamics in the context of drosophila development. While the presence of an INR is convincingly linked to the three-state model, how the INR leads to this additional state is not clear (see first major point below).

Major points:

- 1.1) While the conclusions regarding the link between the presence of an INR and a third state of promoter activity is convincing, the part of the manuscript addressing the pausing itself is less so. Fig.6C' shows that the presence of an INR increases the pausing index, however, this pausing index is the same for the *sna*+INR and the *Kr*-INR1 constructs. The *sna*+INR is better fitted by a three-state model, and the *Kr*-INR1 by a two state model. Thus the pausing itself does not seem correlated to the three state model, which seems at odds with their model in Fig.6B'.

We thank the reviewer for pointing out this issue. Indeed, while the original Pol II ChIP-qPCR results showed clear differences between *sna/sna*+INR and the *kr/kr*-INR pairs, they also display comparable pausing indexes between *sna*+INR and *kr*-INR genotypes. During the revisions process, also prompted by point 1.3, we performed new Pol II ChIP and refined our primer/amplicon selections (see Methods). This optimization significantly improved the ChIP-qPCR results and revealed a clearer difference in pausing indices between *sna*+INR and *kr*-INR. We note however that the pausing index is a proxy for pausing and this index may vary due to other processes than pausing strictly speaking (change in polymerase processivity or elongation rates in the gene body or the promoter-proximal segment, etc...). Moreover, the *sna*+INR and the *kr*-INR promoters differ by more sequence than the INR motif alone¹. Indeed, *sna* possess a TATA box, which is retained in the *sna*+INR synthetic promoter. The presence of a TATA in *sna*+INR (and its absence in *kr*-INR) might also influence the pausing index.

To obtain a clearer picture, we analysed two new genotypes (*Ilp4* and *Ilp4*-INR), in which the effect of the INR cannot be obscured by that of a TATA box. In agreement with the rest of the study, transcription from the paused *Ilp4* promoter is captured by three promoter state while the less stably paused *Ilp4*-INR promoter can be captured by a simple two-state model. The results are shown in Figure 6B and the new Supplemental Figure 8.

In order to investigate if pausing itself was correlated to the three state model, we attempted to affect pausing *in trans* by decreasing the level of NELF complex members. We now show that in *Nelf-A* RNAi embryos, the *kr* promoter switches to two promoter states while it's normally captured by a three-state model in controls. This is shown in

Figure 6C and in Supplemental Figure 9. The disappearance of long waiting times in the *Nelf-A* maternally-depleted embryos further shows that the pausing state is long-lived.

Collectively, we believe that these additional data better link polymerase pausing to the three state model. We acknowledge however that this is one plausible interpretation amongst others.

- 1.2) There is genome-wide data on pausing in drosophila - this data should be cited and/or reanalysed to shed light on the role of the INR in pausing.

We agree that the literature associating the presence of an INR sequence to pausing is quite extensive, particularly in *Drosophila*. We now cite thoroughly some of these in line 289.

- 1.3) The data to back up the pausing analysis is too light. ChIP-qPCR data presented in Fig.6C' should show all data points, and more detailed data should be shown in the supplement (genome tracks, quantification of Pol II density). The authors used a t-test for this data; this is not appropriate since the normality of the distributions cannot be assessed with only 3 replicates.

We apologize if our description of the data presented in former Figure 6C was unclear. We did not perform ChIP-seq (as we are only interested in our synthetic promoter platform) and therefore are unable to show neither genome tracks nor Pol II densities. Instead, we performed Pol II ChIP-qPCR on transgenic embryos, and used specific primers to investigate the enrichment at our synthetic promoters, in comparisons to the body of the reporter gene (yellow), and a gene desert region. We now show in the main figure the three biological replicates and provide for the referee the results, as % of input, for the various genomic locations tested (**Figure 1 for Reviewers**).

Figure 1 for Reviewers: Percent enrichment from Pol II ChIP-qPCR experiments performed on embryonic extracts from the *sna* promoter transgene (A) and the *sna+INR* promoter transgene (B), shown as an average of 3 biological replicates conducted in technical triplicate.

Concerning the statistical test of our Pol II ChIP-qPCR results, as recommended by the referee, we used a Mann Whitney U test that does not assume a normal distribution and modified the figure and text accordingly.

In order to back up the pausing analysis, we performed two additional experiments. First we performed NELF-E ChIP-qPCR on 4 genotypes. As this antibody was a kind gift (Zeitlinger lab²), its quantity was limited and this prevented us from obtaining three biological replicates for all 4 genotypes. The data are therefore presented in Supplemental Figure 9 without a statistical test. However, the trend is similar to what was obtained with Pol II ChIP.

Second, to confirm the link between the third state and pausing, we examined transcription dynamics using RNAi-mediated knockdown against *Nelf-A*. We tested knockdown of maternally supplied NELF-A, which led to a shift from a three- to a two promoter state fitting for the *kr-MS2* transgene. These results are shown in Figure 6C and Supplemental Figure 9 (see also response to point 1.1).

- 1.4) I appreciate that the authors performed calibration of their MS2 signal on RNA FISH, but there is no mention of the actual sensitivity of their live MS2 measurements. What is the minimal number of mRNAs at the transcription spot they require to see a fluorescent spot ? If that threshold is $\gg 1$, how does this affect their inferences of on-off switching, in particular for the strains with very low transcriptional activity ?

The minimal number of mRNA that needs to be present at the transcription site to be detected with our MS2/MCP system and current imaging/calibration settings is in the range of 3-4 transcripts. A sensitivity in the same range ($6^{+/-3}$ nascent mRNA) has also been reported by other labs using MS2/MCP technology in the early fly embryo (e.g. Garcia's lab³). This is now mentioned in the methods section line 838. These numbers are much smaller than the steady-state intensities even of weakly active promoters: the average number at steady state for the *snaTATA*light=18 Pol II; *kr-TATA*=16 Pol II, *llp4*=14 Pol II, *llp4-INR*: 10).

The question on the effect of this calibration procedure on the inference of on-off switching rates is interesting and will be rigorously tested in a separate manuscript, dedicated to the development of an open access software to execute the deconvolution method described in Tantale et al., 2020.

Other points:

- 1.5) What are the error ranges for Fig.4J-L, Fig. 6D and 6J ? Some statistics are needed here

We thank the referee for pointing out this issue. The uncertainty interval can be found in Supplemental Table 2, shown as the minimum and maximum values. We now included 'error bars' in Figure 4J-K corresponding to these uncertainty intervals. We would like to note that there is a difference between uncertainty intervals and confidence interval. Uncertainty intervals are refined bounds corresponding to suboptimal solutions and have no direct statistical meaning. The revised Figure 6 does not show parameter estimates, but all relevant data can be found in Supplemental Table 2 and Supplemental Figure 5.

- 1.6) In the text of Fig.1F, the authors mention that brk goes up to 17% of activation at the end of nc14. However from the plots it seems to reach about 25%.

Thanks for pointing out this mistake; indeed the correct value of activation is 24%. This is corrected in the main text line 144.

- 1.7) In Supplemental Figure 1, the x-axis for the CAGE data should be labelled, as well as the y-axis of the TBP ChIP-seq data.

Supplemental Figure 1 has been modified accordingly.

- 1.8) I suggest avoiding subjective wording such as "incredibly" (second half of page 10).

This has been modified.

- 1.9) I suggest to briefly mention what the yellow reporter gene is - will be helpful for non-drosophilists

A sentence has been included in the main text on line 122.

REVIEWER 2

Dejean et al implemented the MS2/MCP live imaging system and a machine-learning based Markovian promoter state transition model of transcription to characterize how promoter composition dictates transcriptional dynamics in *Drosophila* embryos. They studied two promoter motifs – the TATA box and the INR. The reporter gene driven by the same *snail* minimal enhancer produces distinct transcriptional activities upon various promoter compositions. TATA box-containing promoters (e.g. *snail* promoter) exhibit long active states with high Pol II firing rates and short inactive states. Transcription from the TATA box-

containing promoters can be described with a simple two-state model. On the other hand, the INR-containing promoters demonstrate three-states, with two inactive and one active state. They suggested that the second inactive state is associated with the promoter-proximal pausing, exhibiting inactive period of a few minutes. Lastly, they showed that not all polymerases pause, but only a subset of them enters a paused state.

The authors present interesting and noteworthy results that investigate the role of different promoter motifs in transcriptional regulation. This highly quantitative work is significant to the field of transcriptional regulation and the use of mathematical modeling helps figure out the implication of their experimental observations. **Yet, I believe that some of their findings do not necessarily support their interpretations and conclusions.** For example, while the INR motif supposedly increases the pausing, more transcripts were produced when INR motif was added to the *snail* promoter.

More major comments are mentioned below. Overall, I believe that this work is significant to the field, but the manuscript will need a revision and some clarifications before being published in Nature Communications.

Major Comments:

- 2.1) According to the previous work from Dr. Lagha, paused promoters facilitate synchronous transcriptional activity. In this work, however, the authors seem to say otherwise, where pausing disrupts the synchronous activity (as seen in delayed activation in INR-containing *Kr* and *Ilp4* promoters). Or as shown in Figure 5, pausing-associated INR motif does not affect the synchrony. Can the authors clarify this?

We thank the reviewer for raising this point and apologize if our original formulation led to this confusion. Indeed, in a previous work based solely on the analysis of fixed samples (where timing is artificially reconstituted), we found a correlation between pausing and the degree of synchrony⁴. In the present manuscript, in which we directly quantify synchrony by live imaging, we obtain various scenarios: in some cases differentially paused promoters (as *kr* and *kr*-INR) exhibit similar synchrony profiles, while in other cases, paused promoters such as *Ilp4* lose synchrony when pausing is disrupted (as shown in the new Supplemental Figure 8). Moreover, we have already observed that the non-paused promoter *WntD* leads to synchronous activation, discussed in Ferraro *et al.*, 2016.

Taken together these results do not show that pausing disrupts or fosters synchrony, but rather expands the picture obtained previously. Thus our results nuance the conclusions from Lagha *et al* 2013 and add an extra layer of control to synchrony, namely promoter sequence motifs. Indeed we believe that a non-paused/less stably paused promoter with a strong TATA box (e.g. *WntD*)⁵ is equally able to achieve synchronous expression as a stably paused promoter. Moreover, the classification of pausing status needs to be taken with

care, as recent data point to the high turnover of Pol II, even at highly paused genes such as Hsp70^{2,6}.

In conclusion, we believe there are various ways to achieve synchrony; pausing is one of them, but a strong TATA box is another one.

- 2.2) Similarly, the modeling framework calculates Ton/Toff and Pon/Poff after the initiation of transcriptional activity in each nucleus. Then the model does not describe the delay in transcriptional activation seen in different promoters. Do TATA-box and INR motifs regulate the initiation of transcription differentially? Or is the difference due to other parts of the promoter sequence?

Here the reviewer questions how promoter sequence impacts the time delay between mitosis and initiation. This is an interesting point and indeed, in the original manuscript we refrained from analysing this time window. Instead, the deconvolution pipeline and mathematical framework are used for ‘stationary regimes’, highlighted in grey in Figure 2A.

The analysis of the lag time between mitosis and first initiation is ‘non-stationary’ and was previously modeled using a different modeling paradigm, based on mixed gamma distributions for the time from mitosis to transcriptional activation in nc14^{7,8}. This analysis estimates two parameters, the number of promoter states (a) and transition times between them (b), found to be homogeneous. This modelling paradigm is compatible with the multiple exponential distributions used for modelling successive transcription events at stationarity. The mixed gamma distribution constitutes a special/limiting case of the multiple exponential distributions, when exponential time scales are equal to one another (see Methods, section ‘distribution of the post-mitotic gap’).

To answer to the referee’s comment, we applied this modeling framework to our data by only focusing on the distribution of waiting times between mitosis and first transcriptional activation (illustrated in Supplemental Figure 10A). Interestingly, we found that regardless of the promoter sequence, this distribution can be fitted by a model in which the transition times between states are homogeneously distributed. This is in contrast with the transition times between states in the stationary bursting regime, which are heterogeneous. This result suggests that the states and transitions involved in the two regimes (from mitosis to first activation and at steady-state) are separate and have distinct regulation. In a previous study⁷, we showed that the delay in transcription activation is dependent on the enhancer sequence. These findings go in the same direction.

We thank the reviewer for this comment, which led us to perform a new analysis. We have now included this result as Supplemental Figure 10 and amended the text accordingly (line 430).

- 2.3) The authors showed that the promoter with strong TATA mutation showed similar k_{ini} (Pol II loading rate), and reduced T_{ON} . As a result, the burst size is diminished in *snaTATAlight* and *snaTATAmut* embryos. Then, with comparable Pol II firing rate, shouldn't the average amplitude be similar among three conditions? In other words, each burst would be longer (hence, larger size), but the amplitude per burst should be similar. This doesn't seem to be the case in Fig 3C and in 4A-C.

The referee argues that the amplitude of a burst (contribution of the ON period to the signal) is given by k_{ini} (firing frequency). That is true if T_{ON} is larger than the elongation time for the full sequence (roughly 150s in our case). A smaller T_{ON} leads to smaller burst amplitude, for the same k_{ini} . Furthermore, several bursts can contribute to the intensity amplitude if there is overlap, i.e. if T_{OFF} is smaller than 150s. These arguments suggest that the comparison may be less straightforward in general. For the particular situation pinpointed by the referee, k_{ini} is larger for *sna* compared to *snaTATAlight* and *snaTATAmut* embryos (0.113 compared to 0.076 and 0.077, respectively, see Supplemental Table 2). Furthermore, T_{OFF} is smaller for *sna* leading to increased burst overlap. These two features explain why the amplitude is larger in *sna* embryos. The difference between *snaTATAlight* and *snaTATAmut* is explained by the T_{ON} difference, because T_{ON} of one of these two embryos is smaller than 150s. *snaTATAmut* has a T_{ON} smaller than 150s, hence a smaller burst amplitude.

- 2.4) Based on Figure 5B and C, INR motif does not seem to affect the rate of transcription nor the synchrony, whereas the TATA box motif does change both (Fig 3B and C). If pausing is not responsible for the rate of transcription and the synchrony, what does it do? Does "pausing" simply work to decrease the total mRNA production by having a longer inactive state?

The scope of our study was not to decipher the functions of pausing. We agree with the reviewer interpretation of our data but cannot speculate too much on the roles of pausing. However, contrary to the current view, our data suggest that long pausing is a rare event, and that not all polymerases will enter into a stable long-lasting paused state. As the referee indicates, this rare event could be an extra checkpoint to fine tune the levels of expression, possibly by allowing a long lived OFF state from which initiation cannot occur. Pausing may also alter synchrony, as shown here and previously⁴. In the discussion, we now envisage how this extra step of transcriptional control can be used in complex cell fate transitions and during zygotic genome activation (line 523). Based on the deep quantitative characterization that we undertook in this study, we are currently generating the genetic tools required to assess the exciting question on functions of pausing during ZGA.

- 2.5) In Figure 5C, adding INR to the *snail* promoter and removing INR from the *Kr* promoter both led to increased transcriptional activity. Can the authors explain the observed irony?

We agree with the reviewer that mutation of the INR motif leads to different consequences on transcriptional activity. There is no statistical difference between *sna* and *sna*+INR integral amplitude (Figure 5D), yet there is one between *kr* and *kr*-INR1. As we were surprised by this result, we generated a second mutation of the INR (INR2) and this did not increase transcriptional activity. These are summarized in **Figure 2 for Reviewers** and Supplemental Figure 6C.

If INR causes longer inactive state, why is the integral amplitude from *sna*+INR embryos not different from the *sna* embryos?

We understand that the longer inactive states associated with the presence of an INR would intuitively lead to a decreased transcriptional output (quantified as the integral amplitude). However, the integral amplitude is a convolution of different parameters, particularly the durations of ON and OFF states but also the probability to reach them. Our data suggest that this third state (that we interpret as pausing) occurs with low probability ($p_{\text{pause}}=0.09$ for *sna*+INR, $p_{\text{pause}}=0.15$ for *kr*, $p_{\text{pause}}=0.25$ for *kr*-TATA, $p_{\text{pause}}=0.12$ for *Ilp4*), now mentioned on line 398. These probabilities are summarized in Supplemental Table 2. We have now included them as part of Figure 6E-J and we apologize for not making this low probability clearer in our previous figure.

- 2.6) In Figure 6 and in supplemental figures, the difference between the two-state and the three-state model fit doesn't seem to be that big. In Fig 6A, the two-state model fit is still within the confidence interval. This is also observed in *sna*TATALight+INR fit (Fig S6K). In fact, fit with the three-state model looks better in all cases including the snail promoter. Is this overfitting? If the three-state model works with the promoters without the INR motif,

what do the parameters look like for the “second inactive” state? If the values from the two inactive states are comparable and short, then I think this can be used to show the significance of the “third state” in promoters containing the INR motif.

We apologize if our description of the choice of the best fit was unclear.

Yes, the referee is correct, the three-state fitting will look better – more parameters is *always* better, but using the three-state when the two-state is sufficient indeed leads to overfitting (see below). Per the principle of parsimony, we selected the model that requires the fewest number of states to sufficiently explain our data, which we have clarified in text on line 187 and in Supplemental Table 1.

To determine the number of states in a rigorous way without overfitting, we relied on three complementary features:

a) the confidence interval

The non-parametric (empirical) distribution is estimated with a 95% confidence interval, computed using Greenwood’s formula (as implemented in the MATLAB function `ecdf`).

b) the objective function of the fit

The L2 (Euclidean) distance between the non-parametric and the parametric distributions, actually used for parameter optimization. A parameter α allows setting the scale ($\alpha=1$ for linear, differences of probabilities, $\alpha=0$ for logarithmic, differences of logarithms of probabilities); the estimates in Supplemental Table 1 are given for an intermediate value ($\alpha=0.6$), however they are robust across several α (see methods).

c) a modified Kolmogorov-Smirnov (KS) test

We performed the one-sample KS test using the parametric distribution as reference. The KS test is based on the supremum of the differences in probability and is therefore sensitive to errors in the short time scales and has little sensitivity on rare events. Because our fit is global (includes both long-time scales and very short timescales), the errors are not sufficiently well-compensated on short time scales (less than 10-20 s), and the standard KS test then recommends rejection of both two- and three-state models. In order to distinguish between the two models (two vs three states), we have used only timescales longer than a cut at 10-20 seconds in the KS analysis (the choice was based on the empirical requirement of accepting 3 states models without losing too much information).

We chose to examine all three methods because none is 100% perfect. Among the three methods, the confidence interval method proved to be a robust qualitative indicator, allowing classification of the promoters into two and three states. It has the disadvantage of not being summarized quantitatively by a number. On the quantitative side, the objective function is sensitive to both frequent and rare events but it is difficult to set a separation threshold objectively. The p-value of a KS test is objective, but this

test strongly penalizes errors at very short timescales and asks for a timescale cut, which is an extra parameter, in order to discriminate between models.

The results of the KS test and that of the objective function are now provided in a new table, Supplemental Table 1. The p-value of the KS test is now stated in the figure legend of each 'fitting' panel. For promoters where the decision between a two-state and a three-state is not obvious from the fitted figure (note that the fit are displayed in log-log scale and therefore big difference will look small), such as that for the *snaTATAlight+INR* promoter, the use of this statistical test provides a more objective threshold for model rejection. This new analysis suggests that the *snaTATAlight+INR* genotype, previously classified as a three-state could be considered as a two-state promoter. However we note that this is an exception and all other genotypes of the manuscript can be unambiguously classified as a two- or three-state model, regardless of the statistical method employed. We thank the reviewer for his comment, as it encouraged us to perform a more robust analysis and to revisit the number of states for the *snaTATAlight+INR* genotype. Text and figures have been modified accordingly (line 463-469).

In the case where two promoter states suffice and three promoter states represent overfitting, the estimated values for the parameters of the 'extra OFF' state are highly uncertain and are incompatible with the observed data. We thus cannot use them, as suggested by the referee.

- 2.7) Ton in *Kr*-INR1 embryos is much longer than the Ton from embryos with the *snail* promoter (Fig 6H). Do you think the non-canonical TATA box in the *Kr* promoter drives stronger transcription (hence longer Ton), but the pausing due to INR-motif dampens it?

This is an interesting question that we addressed by creating a synthetic *kr* promoter devoid of any TATA-like sequence (named *kr*-TATA). Without a non-canonical TATA box, the *kr* promoter drives lower levels of expression (Supplemental Figure 6C,G). Similar to the wild-type *kr* promoter *kr*-TATA transcription dynamics can be captured by a three promoter states, with two inactive OFF states. However, the duration of the ON state was severely diminished in *kr*-TATA ($T_{ON}=17\text{sec}$ for *kr*-TATA instead of $T_{ON}=95\text{sec}$ for *kr*). Thus the presence of an INR alone does appear to induce a short ON duration. These results are consistent with the effect of TATA mutation on *sna*.

Interestingly, the probability to occupy the paused state and its duration are higher for *kr*-TATA compared to *kr* (p_{pause} for *kr*-TATA= 0.3; p_{pause} for *kr*=0.02). Thus, our results indeed reveal a potential antagonism between the INR and the non-canonical TATA box in the *kr* promoter. We have included these results in Supplemental Figure 6, and Supplemental Table 2 and amended the text accordingly line 470.

Minor Comments:

- 2.8) Transcriptional activity from each nucleus is variable, and it's even more so in embryos with promoters that drive low transcriptional activity (e.g., *snaTATA*light, *snaTATA*mut) or with INR-motif that has long pausing (e.g., *sna*+INR or *Kr*). In those cases, the average transcriptional activities shown in Figure 1 or 5 do not represent the bursting (or more precisely, the short/long inactive states). Although the authors demonstrated this with a schematic on promoter states, it'll be nice if some representative traces are shown as well (e.g., I used Figure 4A-C to get better ideas of representative transcriptional activity from each construct).

We have added representative traces to Figure 5 as well as Figure 1 to clarify transcriptional activity.

REVIEWER 3

This is a very interesting manuscript by Mounia Lagha's group. The literature is rather vague and anecdotal about the roles of core promoter motifs in transcription dynamics- this is in part because a few disparate studies have made conclusions without fully looking at the complexity and redundancy within promoter elements. So, a comparative imaging study, based on minimal promoter fragments in a standardized locus, is very welcome. Another strength of the system is the use of the fly blastoderm, which benefits from the synchrony of measurement and low cellular heterogeneity that could otherwise interfere with clean inferences. In particular, I find the machine learning approach to dealing with MS2 traces very imaginative and innovative. I wish I had thought of doing this.

The data seem very good, and the study is generally clearly written and methodical. The modelling is by and large well explained.

My only substantial concern is that I find the argument linking the 3rd state in the model to a paused state a little weak (bottom of P16 and top of P17). At the moment it seems to be just a fit to a model. Could this not be another inactive promoter state? Can you justify this as a paused state? I can't follow the reasoning here. It might just need tidying up the text for the reader, it might need a better argument/more data. It is hard for me to tell.

We thank referee 3 for his/her positive feedback on our work. The concern regarding our ability to link the third state to a paused state is very legitimate and was also raised by the two other referees. As we also felt this was a weak part, we undertook new experiments (a novel INR mutant and a *Nelf-A* knock-down), not requested, that will hopefully strengthen this linkage. For a detailed explanation, please refer to answer Referee 1 comment 1.1 and comment 1.3, and to Referee 2 point 2.7 and point 2.4.

References

1. Qi, Z. *et al.* Large-scale analysis of Drosophila core promoter function using synthetic promoters. *bioRxiv* (2019) doi:10.1101/2020.10.15.339325.
2. Shao, W. & Zeitlinger, J. Paused RNA polymerase II inhibits new transcriptional initiation. *Nat. Genet.* **49**, 1045–1051 (2017).
3. Garcia, H. G., Tikhonov, M., Lin, A. & Gregor, T. Quantitative Imaging of Transcription in Living Drosophila Embryos Links Polymerase Activity to Patterning. *Curr. Biol.* **23**, 2140–2145 (2013).
4. Lagha, M. *et al.* Paused Pol II coordinates tissue morphogenesis in the Drosophila embryo. *Cell* **153**, 976–87 (2013).
5. Ferraro, T. *et al.* Transcriptional Memory in the Drosophila Embryo. *Curr. Biol.* **26**, 212–218 (2016).
6. Krebs, A. R. *et al.* Genome-wide Single-Molecule Footprinting Reveals High RNA Polymerase II Turnover at Paused Promoters. *Mol. Cell* **67**, 411-422.e4 (2017).
7. Dufourt, J. *et al.* Temporal control of gene expression by the pioneer factor Zelda through transient interactions in hubs. *Nat. Commun.* **9**, 5194 (2018).
8. Bellec, M., Radulescu, O. & Lagha, M. Remembering the past: Mitotic bookmarking in a developing embryo. *Curr. Opin. Syst. Biol.* **11**, 41–49 (2018).

Reviewers' Comments:

Reviewer #1:

Remarks to the Author:

The authors have adequately addressed the points I raised in the first round of reviews, and I believe that the manuscript is ready for publication. Congratulations for this very nice piece of work !

Reviewer #2:

Remarks to the Author:

The revised manuscript and the letter to the reviewers addressed all the concerns I had raised in the initial submission. I recommend publication of the manuscript.

Reviewer #3:

Remarks to the Author:

Minor comments only. The manuscript is greatly improved and my reservations about the matching between the 3rd state and pausing are gone. The paper is ready for publication.

Minor comments (optional)

Line 269-271: this is also consistent with a reduction of TBP recruitment, if one considers that TBP may be continuously turning over.

275-277. The Dicty mutations were in the context of a native promoter, so there may be other elements compensating for the TATA mutations. The current argument made by the current authors is also feasible- although act5 is also induced a little during early development, as well as being "housekeeping".

REVIEWERS' COMMENTS

Reviewer #1 (Remarks to the Author):

The authors have adequately addressed the points I raised in the first round of reviews, and I believe that the manuscript is ready for publication. Congratulations for this very nice piece of work !

Reviewer #2 (Remarks to the Author):

The revised manuscript and the letter to the reviewers addressed all the concerns I had raised in the initial submission. I recommend publication of the manuscript.

Reviewer #3 (Remarks to the Author):

Minor comments only. The manuscript is greatly improved and my reservations about the matching between the 3rd state and pausing are gone. The paper is ready for publication.

Minor comments (optional)

3.1 Line 269-271: this is also consistent with a reduction of TBP recruitment, if one considers that TBP may be continuously turning over.

We agree with Reviewer #3 that this is also a plausible explanation, and have updated the text to reflect this (line 267-268).

3.2 275-277. The Dicty mutations were in the context of a native promoter, so there may be other elements compensating for the TATA mutations. The current argument made by the current authors is also feasible- although act5 is also induced a little during early development, as well as being “housekeeping”.

Reviewer #3 makes a reasonable observation which we agree with and have added to the text of the manuscript (lines 269-273).